# STEER A CROWD: LEARNING TO PERSUADE A POPULATION IN A STACKELBERG GAME

## ABSTRACT

Multi-agent systems are prevalent across various domains, characterized by misaligned objectives and information asymmetry, which facilitate the study of incentive design and information design. Existing research often assumes known models and static environments. Motivated by this, we propose a Dynamic Incentive and Information Design (DIID) framework for finite-horizon Markov games, involving a principal and multiple agents. Our focus is on how the principal learns her optimal policy based on data generated through interactions with agents. The main challenge lies in balancing the principal's regret and violations of agents' incentive compatibility constraints during interactions. We establish a lower bound characterizing the trade-off between the two objectives and propose an algorithm attaining the optimal trade-off, i.e. $\tilde{\mathcal{O}}(T^{2/3})$ regret and constraint violation. Additionally, with access to additional unilateral deviation information of the agents, we propose an algorithm attaining improved guarantees that achieve $\tilde{\mathcal{O}}(T^{1/2})$ for both regret and constraint violation simultaneously.

## 1 INTRODUCTION

Multi-agent systems, where multiple decision-makers or "agents" interact, have become ubiquitous. These range from drivers selecting routes in traffic networks to financial entities reacting to market signals. Misaligned objectives and information asymmetry in multi-agent systems facilitate characterizing rational behavior and designing incentives and information. These issues have been extensively studied in economic literature. For instance, traffic congestion can be alleviated by steering drivers' route choices. Measures like congestion pricing impose higher fees during peak hours (Barrera & Garcia, 2014), while real-time traffic updates empower drivers to make informed route decisions (Arnott et al., 1991). Similarly, in financial markets, regulators utilize incentives like tax benefits and penalties to guide market behavior (Goodhart et al., 2012), while transparency mechanisms shape investor perceptions, fostering stability (Goldstein & Yang, 2017). Many studies in economic literature have examined the existence and definition of equilibrium (Bergemann & Morris, 2013; 2016) and the properties of the solution.

However, studies in the economic literature often assume that the model is known, with few focusing on how to learn equilibrium notions from data, e.g. learn the model through multiple interactions. Moreover, these studies mainly address static cases and do not account for dynamics in the system, such as how decisions made now affect the evolution of the system's state and future utility. Motivated by this, we propose a general Dynamic Incentive and Information Design (DIID) framework encompassing joint incentive and information design in a finite-horizon Markov game, involving a single principal and multiple agents. The principal manipulates agents through (i) misaligned objectives, by taking actions to directly influence agents' rewards (i.e., incentive design), and (ii) information asymmetry, by having access to a random variable that agents cannot see (an external parameter) to send signals and change the agents' beliefs about that variable, which indirectly influence the followers' actions (i.e., information design). In other words, in our framework, the principal can perform two moves simultaneously in each iteration. It can directly change the agents' reward functions by taking an action, and change the posterior by sending a signal.

Given principal's actions and signaling schemes, in each step, the agents' decision making problem is reduced to a Bayesian game. In this case, the policies of rational agents converge to a Bayesian Correlated Equilibrium (BCE) (Bergemann & Morris, 2013; 2016). Thus, the principal's problem

is to design an incentive and information structure such that the resulting BCE maximizes the principal's cumulative reward, i.e., to find a Stackelberg equilibrium (von Stackelberg, 1952) of the leader-follower game. That is, the principal's optimal policy characterizes the optimal way she steers the agents in her favor, by changing their reward functions and beliefs.

Our focus is on how to learn the principal's optimal policy in the online learning setting, through the data generated by the interaction between the principal and agents. Unlike prior works on online information design (Gan et al., 2022c; Wu et al., 2022; Bernasconi et al., 2022), we do not assume that the principal has knowledge of the agents' reward function. The principal's optimal policy is defined by constrained optimization, where the constraint ensures that the agents adopt a BCE against the principal's policy. As a result, to find the principal's optimal policy efficiently from data, we need to estimate the agents' reward functions and predict how they respond to the principal.

To accurately estimate the reward functions, the principal needs to incentivize the agents to try all actions, which might not be feasible in general. For example, if some actions are strictly dominated and thus never be executed by the agents. In this case, it is impossible for the principal to estimate the rewards on these actions. To bypass this challenge, we assume that the principal has the power to take actions on behalf of the agents, and she additionally suffers from the violation of the BCE constraint. Under this setting, our goal is to design efficient online learning with both sublinear regret and constraint violation. To this end, we need to (a) explore efficiently so as to obtain an accurate estimate of the BCE constraint, and (b) quantify how the estimation error of the BCE constraint affects the regret. Additionally, our algorithm should be able to handle large state spaces by incorporating function approximations.

**Main results and contributions**  Our contributions are three-fold, detailed as follows.

- First, we prove a regret lower bound revealing an intrinsic tension between principal's regret and agents' BCE constraint violation. Specifically, any algorithm can simultaneously achieve an $\tilde{\mathcal{O}}(T^\alpha)$ regret and an $\tilde{\mathcal{O}}(T^{1-\alpha/2})$ constraint violation, for any $\alpha \in (1/2, 1]$. Here $\tilde{O}(\cdot)$ omits logarithmic factors. In particular, with $\alpha = 1/2$, when the principal attains an $\tilde{\mathcal{O}}(\sqrt{T})$ regret, she suffers from a $\Omega(T^{3/4})$ constraint violation. These two terms at best are balanced by setting $\alpha = 2/3$, which leads to a $\tilde{O}(T^{2/3})$ error in total.

- To demonstrate its tightness, we propose a provably efficient algorithm that attains $\tilde{\mathcal{O}}(T^{2/3})$ regret and constraint violation simultaneously by exploiting the technique of reward-free exploration (Jin et al., 2020a; Zhang et al., 2021).

- Additionally, we show that the intrinsic tension between regret and BCE constraint violation can be relieved only by granting the principal more observations. When the principal additionally has access to the reward-maximizing actions and the optimal reward values, we can design an optimistic algorithm that simultaneously attains a $\tilde{O}(\sqrt{T})$ regret and BCE constraint violation. These two algorithms can readily incorporate linear function approximation.

## 2 RELATED WORKS

The principal-agent problems have been a persistent focus in economic research, which encompasses problems such as incentive design (Sandholm, 2003), information design (Kamenica & Gentzkow, 2011), as well as coordination mechanism design for the generalized principal-agent problems that combine both Myerson (1982); Gan et al. (2022a). These works focus on characterizing outcomes in static environments and addressing computational aspects. In recent years, there has also been growing interest in the computation and learning aspects of sequential principal-agent problems. For example, Zhang & Conitzer (2021); Cacciamani et al. (2023a) studied sequential incentive design, and Celli et al. (2020); Gan et al. (2022b); Wu et al. (2022); Bernasconi et al. (2022); Iyer et al. (2023) studied sequential information design.

The closest works to ours are the line that studies Bayesian persuasion (Kamenica & Gentzkow, 2011) under Markov games (Wu et al., 2022; Gan et al., 2022b; Iyer et al., 2023), where the payoff-relevant state evolves according to a Markov chain. Specifically, Wu et al. (2022) considered a finite-horizon Markov persuasion process, where a single principal seeks to persuade a stream of

myopic agents to maximize his cumulative rewards. However, in the learning aspect, it assumes the absence of knowledge about the transition kernel and presumes that the principal knows the reward function. Gan et al. (2022b); Bernasconi et al. (2023) studied a similar setup with a far-sighted receiver, but their focus is on the computational aspect. In contrast, our focus is on learning the optimal persuasion policy from data. Even in the straightforward scenario where the receiver is myopic, we emphasize that the lack of knowledge regarding the receiver's reward presents a significant challenge for learning. A recent work by Iyer et al. (2023) further considered the case where the agent's belief is endogenous, i.e., instead of assuming the agent knows the state, and only needs to maintain an exogenous belief about the external parameter, they assumed the state is unknown, in which case, agent's belief now depends on the realized history of the Markov chain. In summary, two key differences between our work and these aforementioned related works are that (i) we do not make the assumption that the principal knows the agent's reward function, and (ii) we allow multiple agents that play a noncooperative game.

**Notation**  Throughout this paper, we follow the following notation. For function $f, g : \mathcal{X} \to \mathbb{R}$, we use $\langle f, g \rangle := \sum_{x \in \mathcal{X}} f(x)g(x)$ to denote the inner product of $f$ and $g$. We denote $\Delta(\mathcal{X})$ as all the distribution over the finite set $\mathcal{X}$. For a positive-semidefinite matrix $A \in \mathbb{R}^{d \times d}$ and vector $x \in \mathbb{R}^d$, we use $\|x\|_A$ to denote $\sqrt{x^\top A x}$. We use $[n]$ to denote $\{1, 2, \cdots, n\}$. Meanwhile, all norms $\| \cdot \|$ are $\ell_2$-norms unless otherwise specified.

## 3  PROBLEM SETUP

**Markov Game**  We consider a Markov Game between a principal (system designer) and $I$ agents (individuals in the system). In contrast to a typical MDP where only the agent takes action, in this scenario, both the principal and the agents take actions. The principal can influence the agents by directly taking actions and by influencing their beliefs about external parameters. Specifically, we consider $\mathcal{M} = (H, \mathcal{S}, \Omega, \mathcal{A}, \mathcal{B}; \{P_h\}_{h \in [H]}, \{r_h^i\}_{i \times h \in [I] \times [H]}, \{R_h\}_{h \in [H]}, \{\psi_h\}_{h \in [H]})$, where $H$ is the number of steps, $\mathcal{S}$ is the space of states, $\Omega$ is the space of external parameters, $\mathcal{A}$ is the space of the principal's actions, and $\mathcal{B} := \mathcal{B}_1 \times \cdots \mathcal{B}_I$ is the space of action profiles of $I$ agents (with $\mathcal{B}_i$ being the action space for each agent $i \in [I]$). Here, $r_h^i : \mathcal{S} \times \Omega \times \mathcal{A} \times \mathcal{B} \to \mathbb{R}$ and $R_h : \mathcal{S} \times \Omega \times \mathcal{A} \times \mathcal{B} \to \mathbb{R}$ denote the reward function[1] of agent $i$ and the principal, respectively, and $P_h : \mathcal{S} \times \Omega \times \mathcal{A} \times \mathcal{B} \to \Delta(\mathcal{S})$ is the transition function at step $h$. At each step $h \in [H]$, we let $\omega_h \in \Omega$ be an independent external parameter with $\omega_h \sim \psi_h$, and we denote the principal's action as $a_h \in \mathcal{A}$, and $I$ agents' action profile as $\boldsymbol{b_h} := (b_h^1, \cdots, b_h^I) \in \mathcal{B}$. Then the next state is sampled from $s_{h+1} \sim P_h(s_h, \omega_h, a_h, b_h)$. In addition, at each step, we assume state $s_h$ and the prior distribution of external parameter $\psi_h$ are public to both the principal and $I$ agents, while the realized external parameter $\omega_h$ is private to the principal. In other words, the principal has an information advantage over $I$ agents.

**Incentive and information design**  In the Markov game $\mathcal{M}$ defined above, the principal can influence agents' actions $\{b_h\}_{h \in [H]}$ via both *incentive* and *information* design.

- Choosing action $a_h$ to affect each agent $i$'s expected reward $r_h^i(s_h, \omega_h, a_h, b_h)$. Specifically, at the beginning of the game, the principal decides a policy $\{\pi_h\}_{h \in [H]}$, where $\pi_h : \mathcal{S} \to \Delta(\mathcal{A})$ represents how the principal takes actions.

- Shaping the agents' beliefs about $\omega_h$ via signaling. Specifically, principal commits to a signaling scheme $\{\nu_h\}_{h \in [H]}$ and announces to all the agents, where $\nu_h : \mathcal{S} \times \Omega \times \mathcal{A} \to \Delta(\mathcal{B})$. At each step, the principal samples $\boldsymbol{b_h} \sim \nu_h(\cdot | s_h, \omega_h, a_h)$, and recommends action $b_h^i$ privately to the corresponding agent $i$. Based on the recommended action $b_h^i$ and the signaling scheme $\nu_h$, the $i$-th agent can infer their own distribution of the external parameter using Bayes' rule.

In this context, the signaling space (the space signaling scheme maps to) does not need to be limited to the action space; it can encompass any signaling space. According to the revelation principle

---

[1] At each step, the reward is stochastic, while the reward function represents the expectation of the reward. As long as it does not cause confusion, we use $r_h^i, R_h$ to denote the stochastic reward at each step, and $r_h^i(s_h, \omega_h, a_h, b_h), R_h(s_h, \omega_h, a_h, b_h)$ to denote the reward function.

(Kamenica & Gentzkow, 2011), this is equivalent to directly recommending actions. Notably, if we only consider policy $\pi$, it degenerates into a incentive design game, and when we only consider $\nu$, it is an information design problem. Here the principal can influence the agents in both ways and thus our setting is strictly more challenging.

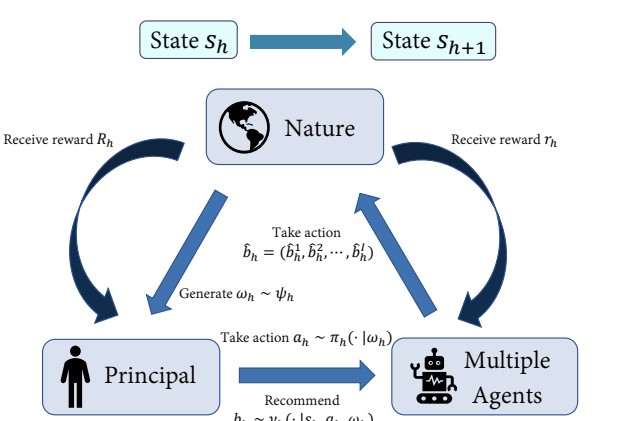 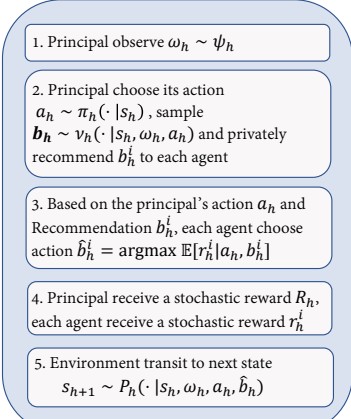

Figure 1: Interaction protocol of the Markov game between the principal and a set of $I$ agents. In each step, the principal first announces the signaling scheme before each episode starts. Then, in each step of the episode, the principal takes an action that enters agents' reward functions, and send a signal in the form of recommended actions to the agents. The agents' posterior belief is altered by the recommended actions. The agents choose actions from the equilibrium by aggregating over the posterior belief. Then both principal and agents receive rewards and transit to a new state.

**Leader-follower structure**  Our game has a leader-follower structure — the principal first announces her policy $\pi$ and signaling scheme $\nu$ and commits to them, and the agents strategically react the principal. Then, with such knowledge, the agents will strategically take their actions in response to the leader. In particular, with multiple agents with diverse reward functions, in each step, the agents essentially play a Bayesian game induced by the principal The goal of the principal is to **steer the agents** such that the outcome of the game is in favor of the principal, i.e., the cumulative rewards of the principal are maximized.

**Bayesian game induced by principal**  In each step $h$, after principal take action $a_h$ and recommend action for agents, $n$ agents play a Bayesian game with reward function $\{r_h^i(s_h, a_h, \bar{\omega}, \bar{\boldsymbol{b}})\}_{i \in [n]}$. Here we use letters with a superscript bar to represent unknown parameters for the agents. For the $i$-th agent, given principal's action $a_h$, principal's signaling scheme $\nu_h$, and principal's private signal $b_h^i$, $i$-th agent's private posterior belief is $\mathbb{P}(\bar{\boldsymbol{b}}^{-i}, \bar{\omega}|s_h, a_h, b_h^i) \propto \psi_h(\bar{\omega})\nu_h(b_h^i, \bar{\boldsymbol{b}}^{-i}|s_h, \bar{\omega}, a_h)$. In this case, each agent seeks to solve $\max_{b^i \in \mathcal{B}^i} \mathbb{E}[r_h^i(s_h, a_h, \bar{\omega}, \bar{\boldsymbol{b}})]$ based on their own belief. By convention of Bayesian persuasion (Kamenica & Gentzkow, 2011), if multiple equilibria exist, agents may choose actions that favor the principal.

**Interaction Protocol**  To summarize, before the game begins, the principal announces an action policy $\{\pi_h\}_{h \in [H]}$ and a signaling scheme $\{\nu_h\}_{h \in [H]}$ and commits to them. At each step $h \in [H]$, the game proceeds as follows

1. Principal first chooses its action $a_h \sim \pi_h(\cdot|s_h)$;

2. Upon observing $\omega_h \sim \psi_h$, the principal samples $\boldsymbol{b_h} \sim \nu_h(\cdot|s_h, \omega_h, a_h)$, and recommends action $b_h^i$ privately to the corresponding agent $i \in [I]$;

3. Based on principal's action $a_h$ and the recommended action $b_h^i$, this situation reduce to a Bayesian game and agents take actions $\hat{\boldsymbol{b}}_h$ from the equilibrium of this Bayesian game.

4. The principal receives a stochastic reward $R_h$, and each agent receives a stochastic reward $r_h^i$; the environment transits to the next state $s_{h+1} \sim P_h(\cdot|s_h, \omega_h, a_h, \hat{b}_h)$.

This setting has significant real-world applications. For instance, consider a scenario where a government (principal) collaborates with electric car manufacturers (agents) to boost EV adoption. The government can influence manufacturers by offering subsidies and tax rebates and by controlling information about future environmental policies and infrastructure plans, such as charging stations. This strategy helps align manufacturers' actions with public health and environmental goals.

**Value function and Bellman equation** For each policy $(\pi, \nu)$, we denote $Q_h^{\pi, \nu}(s, \omega, a, b) :=$ $\mathbb{E}^{\pi, \nu}[\sum_{h'=h}^{H} R_{h'}(s_{h'}, \omega_{h'}, a_{h'}, b_{h'})|s_h = s, \omega_h = \omega, a_h = a, b_h = b]$ as the expected reward starting from step $h$ condition on state, external parameter, action, where the expectation is taken for the randomness of the latent state, policy $(\pi, \nu)$ and the transition kernel. We also define a state function $V_h^{\pi, \nu}(s) := \mathbb{E}^{\pi, \nu}[\sum_{h'=h}^{H} R_{h'}(s_{h'}, \omega_{h'}, a_{h'}, b_{h'})|s_{h'} = s]$. in similar way. The Bellman equation associated with policy pair $(\pi, \nu)$ is $V_h^{\pi, \nu}(s) = \langle Q_h^{\pi, \nu}, \pi_h \otimes \psi_h \otimes \nu_h \rangle_{\Omega \times \mathcal{A} \times \mathcal{B}}(s)$ and $Q_h^{\pi, \nu}(s, \omega, a, b) = R_h(s, \omega, a, b) + P_h V_{h+1}^{\pi, \nu}(s, \omega, a, b)$, where $P_h V_{h+1}^{\pi, \nu}(s, \omega, a, b) = \langle P_h(\cdot|s, \omega, a, b), V_{h+1}^{\pi, \nu}(\cdot) \rangle_{\mathcal{S}}$.

### 3.1 SOLUTION CONCEPT: PRINCIPAL'S OPTIMAL POLICY SUBJECT TO BCE CONSTRAINT

We aim to design a policy pair $(\pi, \nu)$ that satisfy: (i) the principal's expected total return is maximized; (ii) all agents do not have incentive to deviate from the recommended action $\boldsymbol{b_h}$. In the remainder of this section, we will give the definition of BCE and a formal depiction of the problem.

**Bayes Correlated Equilibrium** BCE is the most widely adopted solution concept to describe the stable outcome where no agent has the incentive to deviate, conditional on the knowledge of the recommended action (Bergemann & Morris, 2013; 2016). To formally define BCE, we first introduce the concept of strategy modification: the $i$-th agent can adjust its strategy at each step $h$ through a function $\varphi_h^i : \mathcal{S} \times \mathcal{A} \times \mathcal{B}^i \to \mathcal{B}^i$. The modification is solely based on current information about the underlying environment available to the agent. We define a modified policy $\varphi_h^i \circ \nu_h : \mathcal{S} \times \Omega \times \mathcal{A} \to \Delta(\mathcal{B})$ as $\varphi_h^i \circ \nu_h(\hat{b}^i, b^{-i}|s, \omega, a) = \langle \varphi_h^i(\hat{b}^i|s, a, \cdot), \nu_h(\cdot, b^{-i}|s, \omega, a) \rangle_{\mathcal{B}^i}$. This represents the action distribution when all agents follow $\nu_h$, except for the $i$-th agent, who modifies their action. The BCE constraint is defined as follows.

**Definition 3.1 (BCE Constraint)** *A policy pair $(\pi, \nu)$ is a BCE if for any state $s_h$ with positive visitation measure (with respect to $\pi, \nu$), $\forall \varphi_h^i \in \Phi^i, i \times h \in [I] \times [H]$,*

$$\langle r_h^i, \psi_h \otimes (\varphi_h^i \circ \nu_h - \nu_h) \rangle_{\Omega \times \mathcal{B}}(s_h, a) \leq 0, \forall a \in \mathcal{A} \cap \{a : \pi_h(a|s_h) > 0\}. \tag{1}$$

Note that the $i$-th agent's posterior over the latent space and other agents' actions can be expressed as $\mathbb{P}(b^{-i}, \omega|s_h, a_h, b_h^i) \propto \psi_h(\omega)\nu_h(b_h^i, b^{-i}|s_h, \omega, a_h)$. Thus, the term: $\langle r_h^i, \psi_h \otimes \nu_h \rangle_{\Omega \times \mathcal{B}}(s_h, a_h)$ is the $i$-th agent's expected reward for following recommendation from their own perspective and $\langle r_h^i, \psi_h \otimes (\varphi_h^i \circ \nu_h - \nu_h) \rangle_{\Omega \times \mathcal{B}}(s_h, a_h)$ denotes the benefit $i$-th agent can obtain from modification $\varphi_h^i$. Thus, Definition 3.1 implies that any agent cannot gain advantage through modification.

**Optimal policy – Stackelberg equilibrium** From some initial state $s_1 \in \mathcal{S}$, the optimal policy $\pi^\star, \nu^\star = \{\pi_h^\star, \nu_h^\star\}_{h \in [H]}$ is defined as the solution to the following bilevel optimization problem:

$$\pi^\star, \nu^\star := \operatorname{argmax}_{\pi, \nu} \quad V_1^{\pi, \nu}(s_1)$$
$$\text{s.t.} \quad \pi, \nu \text{ satisfies BCE Constraint in (1).} \tag{2}$$

Here $(\pi^\star, \nu^\star)$ is the principal's policy that maximizes her cumulative rewards, assuming agents always respond by playing a BCE against her. Therefore, $\pi^\star, \nu^\star$ characterize the optimal policy of the principal. We have the claim that, as long as action space $\mathcal{B}$ is finite, Equation (2) has feasible solutions. The detailed description and proof of this proposition is in Appendix B.

### 3.2 PERFORMANCE METRICS: SUBOPTIMALITY AND CONSTRAINT VIOLATION

Given the definition of the optimal policy, we can define the following two performance metrics.

**Suboptimality** To quantify the disparity between $\pi, \nu$ and the optimal policy $\pi^\star, \nu^\star$ in the objective function, for some initial state $s_1$, we can express suboptimality as follows

$$\text{SubOpt}^{\pi, \nu}(s_1) = \text{ReLU}\left(V_1^{\pi^\star, \nu^\star}(s_1) - V_1^{\pi, \nu}(s_1)\right),$$

where $\text{ReLU}(x) = \max\{0, x\}$. Here, we use the ReLU function to ensure that suboptimality is always positive, as in some cases, if the policy pair $(\pi, \nu)$ does not satisfy the constraints, the expected reward may exceed that of the optimal policy pair defined in Equation (2).

**Constraint violation**   To quantify the extent to which the recommended policy violates the BCE constraint, we define Constraint Violation (CV) as $\text{CV}_h^{i,\pi,\nu}(s_h) = \max_{\varphi_h^i \in \Phi^i} \left\langle r_h^i, \psi_h \otimes \pi_h \otimes \left(\varphi_h^i \circ \nu_h - \nu_h\right)\right\rangle_{\Omega \times \mathcal{A} \times \mathcal{B}}(s_h)$, which measures the maximum expected reward an agent can obtain by deviating from the recommended policy at step $h$. Note that by definition, $\text{CV}_h^{i,\pi,\nu}(s_h) \geq 0$. Thus, the CV under policy pair $(\pi, \nu)$ is

$$\text{CV}^{\pi,\nu}(s_1) = \max_{i \in [I]} \text{CV}^{i,\pi,\nu}(s_1) = \max_{i \in [I]} \sum_{h=1}^H \left\langle \bar{d}_h^{\pi,\nu}(\cdot), \text{CV}_h^{i,\pi,\nu}(\cdot)\right\rangle_{\mathcal{S}},$$

where $\bar{d}_h^{\pi,\nu}(\cdot) = \mathbb{P}(s_h = \cdot | s_1 = s_1)$ is the visitation measure with respect to the policy pair $(\pi, \nu)$, the realization of latent state $\omega$ and the transition kernel $P$. In essence, the constraint violation of the $i$-th agent can be understood as follows: when following policy $\pi$ and $\nu$, there is a hypothetical opportunity for the agent to diverge from the prescribed course of action at each step. The constraint violation of $i$-th agent represents the maximum expected reward that the $i$-th agent could obtain by taking such deviations from the recommended policy.

To see the definition of suboptimality and constraint violation corresponds to the optimal policy defined above, we have claim that $\text{SubOpt}^{\pi,\nu}(s_1) = \text{CV}^{\pi,\nu}(s_1) = 0$ if and only if $\pi, \nu$ is optimal policy defined in Equation (2). The detailed description and proof is in Appendix B.

### 3.3   ONLINE LEARNING UNDER DIID FRAMEWORK

In the following section, we aim to address this problem in an online setting. Specifically, the principal initially does not know the model but interacts iteratively with agents in this Markov game, updating their policy.

**Data collection protocol**   In $t$-th episode, principal commits the policy $\pi^t$ and $\nu^t$, then agents take action recommended by policy $\nu^t$. We allow the principal to gain access to the trajectory $\{s_h, \omega_h, a_h, \boldsymbol{b_h}\}_{h \in [H]}$ as well as the bandit feedbacks $\{R_h, \boldsymbol{r_h}\}_{h \in [H]}$, where $\boldsymbol{r_h}$ is realized agents' reward rather than reward function. Here, we assume that agents take the recommended actions compulsorily. We do not know any prior information, including the agents' reward functions. It is impossible to induce agents to take every action, as some actions may strictly dominate, leading agents to disregard other actions. Additionally, certain actions may be difficult to induce, making it costly to explore the entire policy space. Consequently, there is a possibility that specific actions may not be explored, making it impossible to estimate the reward function at this point. This situation could lead to missing out on the best possible action for the principal. A similar assumption is made in Bernasconi et al. (2022).

**Learning performance metric**   We denote $(\pi^t, \nu^t)$ as the policy generated by some online learning algorithm in episode $t$. The regret and the constraint violation of the learning process are defined as $\text{Reg}(T) = \sum_{t \in [T]} \text{SubOpt}^{\pi^t, \nu^t}(s_1^t)$, $\text{CV}(T) = \sum_{t \in [T]} \text{CV}^{\pi^t, \nu^t}(s_1^t)$. If both regret and constraint violation are $o(T)$, the average policy is approximately optimal for the principal.

## 4   INCENTIVE AND INFORMATION DESIGN WITH BANDIT FEEDBACK

Now, a natural research question arise:

*Can we design an algorithm with $\tilde{\mathcal{O}}(\sqrt{T})$ regret and constraint violation simultaneously?*

The challenge of this question is twofold. First, the impact of estimation errors in the BCE constraint on regret remains unclear. As shown later in this bilevel optimization problem, even small errors in the constraints can lead to significantly different in principal's cumulative reward, making it necessary to explicitly estimate the BCE constraint. Second, directly substituting estimated values into the BCE constraint to compute the optimal policy may not adequately address potential constraint violations or regret in a theoretical framework.

## 4.1 Trade-off between Regret and Constraint Violation

The answer of the question we proposed is negative, as there is a fundamental trade-off between regret and constraint violation.

**Theorem 4.1 (Trade-off between Regret and CV)** *For model $\mathcal{M} = (\mathcal{S}, \Omega, \mathcal{A}, \mathcal{B}, P, R, \{r^i\}_{i \in [I]})$ and any online learning algorithm $\mathscr{A} : \mathcal{H}_t \to \Pi \times N$ with bandit feedback, where $\mathcal{H}_t$ represents the set of all feedback before episode t, $\Pi$ and $N$ denote the spaces of policies $\pi$ and $\nu$, respectively. $\pi^t$ and $\nu^t$ are policies based on the algorithm and the model in episode t. We can establish the following trade-off: For any $\alpha \in [1/2, 1]$, there exists $\delta \in (0, 1)$ such that no algorithm $\mathscr{A}$ can simultaneously achieve better than*

$$\text{Reg}(T) = \tilde{\mathcal{O}}(T^\alpha), \quad \text{CV}(T) = \tilde{\mathcal{O}}(T^{1-\alpha/2}).$$

*for any model $\mathcal{M}$ with probability at least $1 - \delta$. Thus, for any algorithm in online learning with bandit feedback, it is impossible to achieve $\tilde{\mathcal{O}}(\sqrt{T})$ regret and constraint violation simultaneously.*

This outcome suggests a fundamental trade-off between regret and constraint violation. If we set $\alpha = 1/2$, the regret is $\tilde{\mathcal{O}}(\sqrt{T})$, but the constraint violation is $\Omega(T^{3/4})$. To attain a good balance between regret and constrained violation, we can set $\alpha = 2/3$, so that the regret and constraint violation are both $\tilde{\mathcal{O}}(T^{2/3})$ according to Theorem 4.1. The detailed proof of Theorem 4.1 is shown in Appendix C, where we construct hard instances as follows.

**A hard instance** The regret lower bound above is established by constructing a hard instance with $H = 1$, and $\mathcal{S}$ being a singleton. The game consists of a principal and two agents, i.e., $I = 2$. Let the two agents be denoted by Agent-a and Agent-b, and their actions are denoted with subscripts 1 and 2, respectively. That is, the agents play a matrix game induced by the principal. We can create two game instances $X$ and $Y$ satisfying the following property:

> The reward functions of the principal and agents in these two game instances are close, but the corresponding equilibria are different.

With this property, finding the optimal policy is challenging, even when the reward functions are estimated accurately. As a result, even a small estimation error in the reward functions can result in a significant error in either regret or constraint violation.

Specifically, let $\mathcal{A}$ be a singleton, and assume the reward functions do not involve the external parameter $\omega_1$. Thus, the reward received by the principal is completely determined by the BCE of the matrix game of the agents. We assume the rewards of the agents are Bernoulli random variables, whose expectations are listed in Table 1.

Table 1: Pair of hard instances X (left) and Y (right)

| $r_1, r_2, R$ | $b_1$ | $b_2$ | | $r_1, r_2, R$ | $b_1$ | $b_2$ |
|---|---|---|---|---|---|---|
| $a_1$ | $1, \frac{1}{2}, 1$ | $0, \frac{1+\varepsilon}{2}, 0$ | | $a_1$ | $1, \frac{1}{2}, 1$ | $0, \frac{1-\varepsilon}{2}, 0$ |
| $a_2$ | $1, \frac{1}{2}, 1$ | $0, \frac{1+\varepsilon}{2}, 0$ | | $a_2$ | $1, \frac{1}{2}, 1$ | $0, \frac{1-\varepsilon}{2}, 0$ |

Specifically, in instances $X$ and $Y$, under BCE, Agent-b always chooses action $b_2$ and $b_1$ respectively. As a result, the principal receives rewards zero and one respectively. In summary, in our construction, whenever we fail to find the correct BCE, we incur a constant regret and $\varepsilon$ constraint violation. Finding the correct BCE is hard because the reward functions of $X$ and $Y$ are close. See Appendix C for a detailed proof of the regret lower bound. Similar observations have been made in related problems (Bernasconi et al., 2022; Cacciamani et al., 2023b).

## 4.2 Algorithm Design and Theoretical Guarantee

To illustrate the tightness of the lower bound result in Theorem 4.1 with $\alpha = 2/3$, wherein both regret and constraint violation exhibit an optimal asymptotic growth of $\tilde{\mathcal{O}}(T^{2/3})$, we propose an explore-then-commit type algorithm.

To resolve the challenge we mentioned in Section 4, exploration of the unknown reward functions of the principal and agents is needed. The most ideal exploration method is an algorithm that achieves $\tilde{\mathcal{O}}(\sqrt{T})$ regret, such as the optimism principle (Auer et al., 2002). However, the absence of trajectories or reward information under $(\pi, \hat{\varphi}^i \circ \nu)$ poses a technical challenge for us. Specifically, We are unable to handle the expected estimation error of $r_h^i$ under policy $\pi$ and $\hat{\varphi}^i \circ \nu$ in the constraint violation analysis. Therefore, we take a step back and aim to uniformly explore the reward function at all points. Consequently, we propose an explore-then-commit method, as illustrated in Algorithm 1. Initially, we conduct reward-free exploration (Wang et al., 2020; Jin et al., 2020a; Kong et al., 2023) to collect data $\mathcal{D}$ (line 3). Then we use data $\mathcal{D}$ to estimate the unknown parameters and solve the bilevel optimization to compute the policy (line 5-10), which we commit to (line 11-13). We refer to these three phases as the exploration, planning, and commitment phases.

---

**Algorithm 1** Algorithm for bandit feedback

---

1: **Input**: Failure probability $\delta$
2: **for** $t = 1, 2, \cdots, K$ **do**
3:     Run reward-free exploration algorithm to collect data $\mathcal{D}$
4: **end for**
5: $\tilde{Q}_{H+1}(\cdot, \cdot, \cdot, \cdot) \leftarrow 0$ and $\tilde{V}_{H+1}(\cdot) = 0$
6: **for** $h = H, H-1, \cdots, 1$ **do**
7:     Update principal's $Q$−function $\tilde{Q}_h$ and agents' $r$−function $\{\tilde{r}_h^i\}_{i \in [I]}$
8:     Solve optimization problem defined in Equation (4) to get policy $\tilde{\pi}_h, \tilde{\nu}_h$
9:     $\tilde{V}_h(\cdot) = \langle \tilde{Q}_h, \tilde{\psi}_h \otimes \tilde{\pi}_h \otimes \tilde{\nu}_h \rangle(\cdot)$
10: **end for**
11: **for** $t = K+1, K+2, \cdots, T$ **do**
12:     Exploit policy $\{\tilde{\pi}_h, \tilde{\nu}_h\}_{h \in [H]}$
13: **end for**

---

**Exploration phase** As the external parameter $\omega_h$ is independently sampled from the prior distribution $\psi_h$, we can simply estimate it by empirical distribution:

$$\psi_h^k(\omega) = \begin{cases} \frac{1}{k-1} \sum_{\tau=1}^{k-1} \mathbb{1}_{\{\omega_h^\tau = \omega\}} & k > 1 \\ |\Omega|^{-1} & k = 1 \end{cases} \tag{3}$$

The core idea of the exploration phase using the reward-free exploration method is to use the uncertainty function as the reward function. In a simple tabular setting, The uncertainty function at $(s, a)$ is inversely related to the square root of the number of times the state-action pair has been visited. Thus, we collect data from underexplored state-action pairs. Consequently, after reward-free exploration, we obtain an estimated agents' reward $\tilde{r}_h^i$ and the principal's estimated $Q$-fucntion $\tilde{Q}_h$. And there exist two uncertainty function $\tau_h, \Gamma_h : \mathcal{S} \times \Omega \times \mathcal{A} \times \mathcal{B} \to \mathbb{R}$, such that $0 \leq \tilde{r}_h^i - r_h^i \leq \tau_h$ and $\Gamma_h \leq \delta_h = R_h + P_h \tilde{V}_{h+1} - \tilde{Q}_h \leq 0$ for any $(s, \omega, a, b) \in \mathcal{S} \times \Omega \times \mathcal{A} \times \mathcal{B}$. This uncertainty function depicts the estimation error for agents and principal.

The difference from standard reward-free exploration is that we need to handle the external parameters by taking the expectation of the external parameter using empirical distribution during the greedy step. The detailed reward-free algorithm is shown in Appendix A.1.

**Planning phase** In the planning phase, we use the superscript tilde to denote estimated quantities. We leverage optimism principal Auer et al. (2002) in the estimation of principal's $Q$−function and agents' $r$−function, e.g. UCB-VI (Azar et al., 2017) in the tabular case, LSVI (Jin et al., 2020b) in the linear case. To upper bound the term $\mathbb{E}^{\pi^\star, \nu^\star}[\langle \tilde{Q}_h, \tilde{\psi}_h \otimes \pi_h^\star \otimes \nu_h^\star - \tilde{\psi}_h \otimes \tilde{\pi}_h \otimes \tilde{\nu}_h \rangle(s_h)]$ in the principal's regret, we need the optimal policy defined in Equation (2) to satisfy the estimated constraint set. Thus, we cannot directly plug in all the estimated quantities into the BCE constraint; instead, we relax the constraint set by adding an error term. Specifically, at each step $h$, we solve:

$$\max_{(\pi_h, \nu_h)} \quad \left\langle \tilde{Q}_h, \tilde{\psi}_h \otimes \pi_h \otimes \nu_h \right\rangle(s)$$

$$\text{s.t.} \quad \left\langle \tilde{r}_h^i, \tilde{\psi}_h \otimes \pi_h \otimes (\varphi_h^i \circ \nu_h - \nu_h) \right\rangle(s) \leq \langle \tau_h, \tilde{\psi}_h \otimes \pi_h \otimes \varphi_h^i \circ \nu_h \rangle(s) + 4c_\omega |\Omega|^{1/2}/\sqrt{K},$$

$$\forall \varphi_h^i \in \Phi^i, i \in [I], s \in \mathcal{S},$$

$$\tag{4}$$

where $c_\omega = \sqrt{\ln(12H/\delta)}$. We plug in the estimated terms (denoted by superscript tilde) into the constraint and introduce an error term to relax the constraint set, then solve the optimization problem. Here the term $\langle \tau_h, \tilde{\psi}_h \otimes \pi_h \otimes \varphi_h^i \circ \nu_h \rangle$ arises from the statistical error of $r_h^i$, and $4c_\omega |\Omega|^{\frac{1}{2}}/\sqrt{K}$ originates from the statistical error of the prior $\psi_h$. Then we verify that the constraint set defined in Equation (4) is a relaxation of the true BCE constraint set.

**Lemma 4.2** *With probability at least $1 - \delta/3$, all policies that satisfy BCE Constraint (1) also fulfill the relaxed constraint specified in Equation (4).*

Thus, the optimal policy $\{\pi_h^\star, \nu_h^\star\}_{h \in [H]}$ lies within the constraint in Equation (4), which enables us to upper bound the term $\mathbb{E}^{\pi^\star, \nu^\star}[\langle \tilde{Q}_h, \tilde{\psi}_h \otimes \pi_h^\star \otimes \nu_h^\star - \tilde{\psi}_h \otimes \tilde{\pi}_h \otimes \tilde{\nu}_h \rangle(s_h)]$ in the principal's regret by 0. Meanwhile, Proposition B.1 states the non-emptiness of the true BCE constraint set, which guarantees the optimization problem defined in Equation (4) is solvable. Finally, we can ensure that the expectation of error term on the right-hand side can be controlled through the exploration phase.

Subsequently, we utilize the policy $\{\tilde{\pi}_h, \tilde{\nu}_h\}_{h \in [H]}$ determined in the planning stage for the remainder of the online learning process, then we have the following guarantee.

**Lemma 4.3 (Informal)** *With high probability, we have*

$$\mathrm{SubOpt}^{\tilde{\pi}, \tilde{\nu}}(\tilde{s}_1) + \mathrm{CV}^{i, \tilde{\pi}, \tilde{\nu}}(\tilde{s}_1)$$

$$\leq -\sum_{h \in [H]} \Gamma_h(\tilde{s}_h, \tilde{\omega}_h, \tilde{a}_h, \tilde{b}_h) + \sum_{h \in [H]} \mathbb{E}^{\tilde{\pi}, \tilde{\nu}}\Big[\big\langle \tau_h, \psi_h \otimes \tilde{\pi}_h \otimes (\tilde{\nu}_h + \hat{\varphi}_h^i \circ \tilde{\nu}_h)\big\rangle(s_h)\Big] + \tilde{\mathcal{O}}\big(H^2|\Omega|^{1/2}/\sqrt{K}\big),$$

*where $\{\tilde{s}_h, \tilde{\omega}_h, \tilde{a}_h, \tilde{b}_h\}_{h \in [H]}$ is the trajectory generated by policy $(\tilde{\pi}, \tilde{\nu})$.*

This lemma informally demonstrates that the suboptimality and constraint violation of the policy $(\tilde{\pi}, \tilde{\nu})$ are bounded above by the estimation errors of the principal under $(\tilde{\pi}, \tilde{\nu})$ and the the estimation error of agents under $(\tilde{\pi}, \tilde{\nu})$ and $(\tilde{\pi}, \hat{\varphi}^i \circ \tilde{\nu})$. For ease of presentation, we use the context of Markov Games with linear function approximation, as introduced in the work of Jin et al. (2020b). It is noted that (i) the tabular case is a special case of Markov games with linear function approximation; (ii) the framework does not depend on linear structure, it can go beyond the linear case, e.g., general function approximation (Zhang et al., 2023) by replacing $d$ to Eluder dimension(Jin et al., 2021).

**Markov game with linear function approximation** There exists a known mapping $\phi : \mathcal{S} \times \Omega \times \mathcal{A} \times \mathcal{B} \to \mathbb{R}^d$, so that principal's reward $R_h = \langle \phi(s, \omega, a, b), \Theta_h \rangle$, $i$-th agent's reward $r_h^i = \langle \phi(s, \omega, a, b), \theta_h^i \rangle$, transition kernel $P_h(\cdot|s, \omega, a, b) = \langle \phi(s, \omega, a, b), \mu_h(\cdot) \rangle$, where parameters $\Theta_h \in \mathbb{R}^d$, $\theta_h^i \in \mathbb{R}^d$ are unknown , and $\mu_h(\cdot) = (\mu_h^{(1)}, \mu_h^{(2)}, \cdots, \mu_h^{(d)})$ are unknown measures over $\mathcal{S}$. Without loss of generality, we assume that $\|\phi(s, \omega, a, b)\| \leq 1, \forall(s, \omega, a, b) \in \mathcal{S} \times \Omega \times \mathcal{A} \times \mathcal{B}$, and $\{\|\Theta_h\|, \|\theta_h^i\|, \|\mu_h(s)\|\} \leq \sqrt{d}, \forall s \in \mathcal{S}$. If we let $d = |\mathcal{S}||\Omega||\mathcal{A}||\mathcal{B}|$, each coordinate can be indexed by pair $(s, \omega, a, b)$, and mapping $\phi(s, \omega, a, b) = \mathbb{1}_{(s, \omega, a, b)}$ be the canonical basis in $\mathbb{R}^d$, Markov games with linear function approximation framework recovers tabular Markov games model. The complete version of this algorithm is outlined in Appendix A.2.

**Regret and constraint violation upper bound** Based on Lemma D.1, Lemma D.2 and Lemma E.1, we can establish the following result that upper bounds the regret and constraint violation.

**Theorem 4.4 (Analysis of Algorithm 1)** *The explore-then-commit algorithm has the following guarantee, with probability at least $1 - \delta$:*

$$\sum_{t \in [T]} \left(\mathrm{SubOpt}^{\pi^t, \nu^t}(s_1^t) + \mathrm{CV}^{\pi^t, \nu^t}(s_1^t)\right) = \tilde{\mathcal{O}}\left(\sqrt{H^6 d^3 |\Omega|} \cdot T^{\frac{2}{3}}\right).$$

Theorem 4.4 indicates the tightness of the lower bound result with $\alpha = 2/3$, where the growth of $\tilde{\mathcal{O}}(T^{2/3})$, as stated in Theorem 4.1, is indeed attainable and optimal in $T$. The terms $d^3$ and $H^6$ within the square root are typically encountered in standard reward-free algorithms, which require collecting $\tilde{\mathcal{O}}(H^6 d^3/\varepsilon^2)$ trajectories to achieve an $\varepsilon$-optimal policy. The term $|\Omega|$ arises from estimating the prior $\psi_h$ through empirical observation, reflecting the convergence rate of the empirical distribution. Here we use linear function approximation to handle infinite state space. In the tabular case, using the UCB-VI algorithm (Azar et al., 2017), the other factors can also achieve optimality. The proof of Theorem 4.4 is available in Appendix E.

## 5 INCENTIVE AND INFORMATION DESIGN WITH ADDITIONAL FEEDBACK

When facing the inherent challenge of DIID as discussed in Section 4.1, a natural question arises:

Can the principal do better than $\tilde{\mathcal{O}}(T^{2/3})$ if given access to additional feedback from agents?

We answer this question affirmatively in this section, by proposing an algorithm that simultaneously attains $\tilde{\mathcal{O}}(\sqrt{T})$ regret and constraint violation.

**Data collection protocol** We consider a special feedback structure where, at each step, $i$-th agent reports not only the reward $r_h^i$ that it actually receives but also the action $\hat{b}_h^i$ that optimizes its self-interests after receiving $b_h^i$. Additionally, agents report the reward $r_h^i(s_h, \omega_h, a_h, \hat{b}_h^i, b_h^{-i})$, where agent $i$ alone takes $\hat{b}_h^i$ and others follow the recommended actions.

Gathering extra feedback from agents is a prevalent practice across diverse application settings. Beyond assessing the actions we suggest, we also engage in user research and obtain feedback from platform users. For example, an e-commerce platform (Li et al., 2023) recommends sellers the type of commodity based on various factors, such as market demand and trends. However, given that sellers prioritize their income, the platform recommends commodity types and conducts surveys to understand sellers' preferences and expectations regarding rewards.

**Algorithm design utilizing additional feedback** We use the superscript $t$ to denote estimated quantities. Note that additional feedback allows the use of the optimism principle (Auer et al., 2002) for exploration, where we add a bonus term to encourage exploration while estimating the principal's $Q$-function and agents' rewards. Thanks to additional feedback, we are able to bound the term $\mathbb{E}^{\pi^t, \nu^t}\big[\sum_{h=1}^{H} \langle \tau_h^t, \psi_h \otimes \pi_h^t \otimes \hat{\varphi}_h^{i,t} \circ \nu_h^t \rangle \big]$ by $\mathbb{E}^{\pi^t, \nu^t}\big[\sum_{h=1}^{H} \|\phi(s_h^t, \omega_h^t, a_h^t, \hat{b}_h^{i,t}, b_h^{-i,t})\|_{(\Lambda_h^{i,t})^{-1}}\big]$, where $\{s_h^t, \omega_h^t, a_h^t, b_h^t, \hat{b}_h^t\}_{h \in [H]}$ denotes the trajectory generated by policy $(\pi^t, \nu^t)$. Leveraging additional feedback, we are able to bound it using the elliptical potential lemma (Abbasi-Yadkori et al., 2012).

We maintain the update of the principal's $Q-$function as in Algorithm 1, but leverage all the additional feedback for the estimation of the agent's reward function. Thus, we obtain a better uncertainty quantifier $\tau_h^t$ for each $t \in [T]$, which enables us to solve the optimization problem:

$$\max_{(\pi_h, \nu_h)} \quad \langle Q_h^t, \psi_h^t \otimes \pi_h \otimes \nu_h \rangle (s)$$

$$\text{s.t.} \quad \Big\langle r_h^{i,t}, \psi_h^t \otimes \pi_h \otimes (\varphi_h^i \circ \nu_h - \nu_h) \Big\rangle (s) \leq \langle \tau_h^t, \psi_h^t \otimes \pi_h \otimes \varphi_h^i \circ \nu_h \rangle (s) + 4c_\omega |\Omega|^{1/2}/\sqrt{t},$$

$$\forall \varphi_h^i \in \Phi^i, i \in [I], s \in \mathcal{S}$$

(5)

to get policy pair $(\pi_h^t, \nu_h^t)$. As in Equation (4), the constraint set is relaxed using the confidence bound of the unknown parameters, except that now we have a better uncertainty quantifier, thanks to the additional feedback. Similarly, we can conclude that with high probability, the optimal policy $(\pi^\star, \nu^\star)$ lies within the constraint in Equation (5), which enables us to upper bound the term $\mathbb{E}^{\pi^\star, \nu^\star}[\langle Q_h^t, \psi_h^t \otimes \pi_h^\star \otimes \nu_h^\star - \psi_h^t \otimes \pi_h^t \otimes \nu_h^t \rangle (s_h)]$ in the principal's regret. Then we exploit policy $(\pi^t, \nu^t)$ to get new data. Remarkably, as we are solving linear programming to find the optimal solution, it is straightforward to confirm that the algorithm is efficient. The formal depiction of the algorithm is given in Algorithm 2 (See Appendix A.3, full version in Appendix A.4).

**Regret and constraint violation upper bound** In summary, we have the following bound within framework of Markov games with linear function approximation for additional feedback scenarios.

**Theorem 5.1 (Analysis of Algorithm 2)** *The algorithm for additional feedback setting has the following guarantee, with probability at least $1 - \delta$:*

$$\sum_{t \in [T]} \Big( \text{SubOpt}^{\pi^t, \nu^t}(s_1^t) + \text{CV}^{\pi^t, \nu^t}(s_1^t) \Big) = \tilde{\mathcal{O}}\left( \sqrt{H^4 d^3 |\Omega|} \cdot \sqrt{T} \right)$$

The inclusion of terms $H^4$ and $d^3$ within the square root is a standard feature in linear MDPs. The term $|\Omega|$ arises from estimating the prior $\psi_h$ through empirical observation. Here, we address this problem within the Markov game with linear function approximation framework. However, extending the solution to the more general function approximation framework (Zhang et al., 2023) is practical. The detailed proof can be found in Appendix F.

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

# A  OMIITED DESCRIPTIONS AND ALGORITHMS

## A.1  SUB-ALGORITHM FOR REWARD-FREE EXPLORATION

---

**Sub-Algorithm A.1** Reward-Free Exploration

---

1: $Q_{H+1}^k(\cdot,\cdot,\cdot,\cdot) \leftarrow 0$ and $U_{H+1}^k(\cdot) = 0$
2: **for** $h = H, H-1, \cdots, 1$ **do**
3:     $\Lambda_h^k \leftarrow \sum_{\tau=1}^{k-1} \phi(s_h^\tau, \omega_h^\tau, a_h^\tau, b_h^\tau)\phi(s_h^\tau, \omega_h^\tau, a_h^\tau, b_h^\tau)^\top + \lambda I$
4:     $\psi_h^k$ is defined in Equation (3)
5:     $u_h^k \leftarrow \min\{\beta \cdot \sqrt{\phi^\top (\Lambda_h^k)^{-1}\phi}, H\}$
6:     Define the exploration-driven reward function $z_h^k \leftarrow \frac{1}{H}u_h^k$
7:     $\iota_h^k \leftarrow (\Lambda_h^k)^{-1}\sum_{\tau=1}^{k-1}\phi_h^\tau \cdot U_{h+1}^k(s_{h+1}^\tau)$, where $\phi_h^\tau$ is abbreviation of $\phi(s_h^\tau, \omega_h^\tau, a_h^\tau, b_h^\tau)$
8:     $Q_h^k \leftarrow \min\{\left[(\iota_h^k)^\top \phi + z_h^k + u_h^k\right], H\}$
9:     $U_h^k(s) = \max_{(a,b)\in\mathcal{A}\times\mathcal{B}}\langle Q_h^k, \psi_h^k\rangle_\Omega(s,a,b)$, and $(\hat{a}, \hat{b})$ denotes the maximizers
10:     $\pi_h^k(a|s) = \mathbb{1}_{\{a=\hat{a}\}}, \nu_h^k(b|s,a,\omega) = \mathbb{1}_{\{b=\hat{b}\}}$
11: **end for**
12: Using policy $\pi^k, \nu^k$ to explore the environment and get a new trajectory

---

## A.2  THE FULL VERSION OF ALGORITHM 1

---

**Algorithm A.2** Algorithm for Bandit Feedback (The Full version of Algorithm 1)

---

1: **Input**: Failure probability $\delta > 0$, $\beta = c_\beta dH\sqrt{\log(dK/\delta)}$ and $\gamma = c_\gamma\sqrt{d\log(K/\delta)}$
2: **for** $t = 1, 2, \cdots, K$ **do**
3:     Run reward-free exploration algorithm (see Sub-Algorithm A.1)
4: **end for**
5: $\tilde{Q}_{H+1}(\cdot,\cdot,\cdot,\cdot) \leftarrow 0$ and $\tilde{V}_{H+1}(\cdot) = 0$
6: **for** $h = H, H-1, \cdots, 1$ **do**
7:     $\Lambda_h \leftarrow \sum_{\tau=1}^K \phi(s_h^\tau, \omega_h^\tau, a_h^\tau, b_h^\tau)\phi(s_h^\tau, \omega_h^\tau, a_h^\tau, b_h^\tau)^\top + \Lambda$
8:     $u_h \leftarrow \beta \cdot \sqrt{\phi^\top(\Lambda_h)^{-1}\phi}$
9:     $v_h \leftarrow \gamma \cdot \sqrt{\phi^\top(\Lambda_h)^{-1}\phi}$
10:     $\tilde{\psi}_h(\omega) = \frac{1}{K}\sum_{\tau=1}^K \mathbb{1}_{\omega_h^\tau=\omega}$
11:     $\tilde{\eta}_h \leftarrow (\Lambda_h)^{-1}\sum_{\tau=1}^K \phi_h^\tau \cdot (R_h(s_h^\tau, \omega_h^\tau, a_h^\tau, b_h^\tau) + \tilde{V}_{h+1}(s_{h+1}^\tau))$
12:     $\tilde{Q}_h(\cdot,\cdot,\cdot,\cdot) \leftarrow \min\{\left[(\tilde{\eta}_h)^\top\phi + u_h\right](\cdot,\cdot,\cdot,\cdot), H\}$
13:     **for** $i = 1, 2, \cdots, I$ **do**
14:         $\tilde{\theta}_h^i \leftarrow (\Lambda_h)^{-1}\sum_{\tau=1}^K \phi_h^\tau \cdot r_h^i(s_h^\tau, \omega_h^\tau, a_h^\tau, b_h^\tau)$
15:         $\tilde{r}_h^i(\cdot,\cdot,\cdot,\cdot) = \min\{\left[(\tilde{\theta}_h^i)^\top\phi + v_h\right](\cdot,\cdot,\cdot,\cdot), 1\}$
16:     **end for**
17:     For any $s \in \mathcal{S}$, solving the problem defined in Equation (4) to obtain $\tilde{\pi}_h, \tilde{\nu}_h$
18:     $\tilde{V}_h(\cdot) = \langle \tilde{Q}_h, \tilde{\psi}_h \otimes \pi_h \otimes \nu_h\rangle(\cdot)$
19: **end for**
20: **for** $t = K+1, K+2, \cdots, T$ **do**
21:     Exploit policy $\{(\pi_h, \nu_h)\}_{h\in[H]}$
22: **end for**

---

### A.3 Algorithm for additional feedback

---

**Algorithm 2** Algorithm for additional feedback

---

1: **Input**: Failure probability $\delta$
2: **for** $t = 1, 2, \cdots T$ **do**
3: $\quad Q_{H+1}^t(\cdot, \cdot, \cdot, \cdot) \leftarrow 0$ and $V_{H+1}^t(\cdot) = 0$
4: $\quad$ **for** $h = H, H-1, \cdots, 1$ **do**
5: $\quad\quad \psi_h^t$ is defined in Equation (3)
6: $\quad\quad$ Update principal's $Q-$function $Q_h^t$ and agents' $r-$function$\{r_h^{i,t}\}_{i \in [I]}$
7: $\quad\quad$ Solve optimization problem defined in Equation (5) to get policy $\pi_h^t, \nu_h^t$
8: $\quad\quad V_h^t(\cdot) = \langle Q_h^t, \psi_h^t \otimes \pi_h^t \otimes \nu_h^t \rangle(\cdot)$
9: $\quad$ **end for**
10: $\quad$ Exploit policy $\{\pi_h^t, \nu_h^t\}_{h \in [H]}$ to get new data
11: **end for**

---

### A.4 The Full version of Algorithm 2

---

**Algorithm A.4** Algorithm for Additional Feedback (The Full version of Algorithm 2)

---

1: **Input**: Failure probability $\delta > 0$, $\beta = c_\beta dH\sqrt{\log(dT/\delta)}$ and $\gamma = c_\gamma\sqrt{d\log(T/\delta)}$
2: **for** $t = 1, 2, \cdots T$ **do**
3: $\quad Q_{H+1}^t(\cdot, \cdot, \cdot, \cdot) \leftarrow 0$ and $V_{H+1}^t(\cdot) = 0$
4: $\quad$ **for** $h = H, H-1, \cdots, 1$ **do**
5: $\quad\quad \Lambda_h^t \leftarrow \sum_{\tau=1}^{t-1} \phi(s_h^\tau, \omega_h^\tau, a_h^\tau, b_h^\tau)\phi(s_h^\tau, \omega_h^\tau, a_h^\tau, b_h^\tau)^\top + \lambda I$
6: $\quad\quad u_h^t \leftarrow \beta \cdot \sqrt{\phi^\top(\Lambda_h^t)^{-1}\phi}$
7: $\quad\quad \psi_h^t$ is defined in Equation (3)
8: $\quad\quad \eta_h^t \leftarrow (\Lambda_h^t)^{-1}\sum_{\tau=1}^{t-1}\phi(s_h^\tau, \omega_h^\tau, a_h^\tau, b_h^\tau) \cdot \left(R_h(s_h^\tau, \omega_h^\tau, a_h^\tau, b_h^\tau) + V_{h+1}^t(s_{h+1}^\tau)\right)$
9: $\quad\quad Q_h^t(\cdot, \cdot, \cdot, \cdot) = \min\{\left[(\eta_h^t)^\top\phi + u_h^t\right](\cdot, \cdot, \cdot, \cdot), H\}$
10: $\quad\quad$ **for** $i = 1, 2, \cdots, I$ **do**
11: $\quad\quad\quad \Lambda_h^{i,t} = \lambda I + \sum_{\tau=1}^{t-1}\phi(s_h^\tau, \omega_h^\tau, a_h^\tau, \hat{b}_h^{i,\tau}, b_h^{-i,\tau})\phi(s_h^\tau, \omega_h^\tau, a_h^\tau, \hat{b}_h^{i,\tau}, b_h^{-i,\tau})^\top$
12: $\quad\quad\quad v_h^{i,t} \leftarrow \gamma \cdot \sqrt{\phi^\top(\Lambda_h^t + \Lambda_h^{i,t})^{-1}\phi}$
13: $\quad\quad\quad \theta_h^{i,t} = (\Lambda_h^t + \Lambda_h^{i,t})^{-1}\sum_{\tau=1}^{t-1}\left(\phi_h^\tau \cdot r_h^{i,\tau} + \hat{\phi}_h^{i,\tau} \cdot \hat{r}_h^{i,\tau}\right)$
14: $\quad\quad\quad r_h^{i,t}(\cdot, \cdot, \cdot, \cdot) = \min\{\left[(\theta_h^{i,t})^\top\phi + v_h^t\right](\cdot, \cdot, \cdot, \cdot), 1\}$
15: $\quad\quad$ **end for**
16: $\quad\quad$ For any $s \in \mathcal{S}$, solving the problem defined in Equation (5) to obtain $\pi_h^t, \nu_h^t$
17: $\quad\quad V_h^t(\cdot) = \langle Q_h^t, \psi_h^t \otimes \pi_h^t \otimes \nu_h^t \rangle(\cdot)$
18: $\quad$ **end for**
19: **end for**

---

## B OMITTED PROOFS FOR PROPOSITION

### B.1 EXISTENCE OF BCE CONSTRAINT

**Proposition B.1** *If the space $\mathcal{B}$ is (i) finite or (ii) infinite with $\mathcal{B}^i$ is a compact Hausdorff space and $r_h^i$ is continuous for any $i \times h \in [I] \times [H]$, there exists a policy pair $(\pi, \nu)$ that satisfies Constraint (1).*

**Proof** According to (Hart & Schmeidler, 1989), for any $s_h \in \mathcal{S}, a_h \in \mathcal{A}, \omega_h \in \Omega, h \in [H]$, and $i \in [I]$ there exist a Correlated Equilibrium $\nu_h(\cdot|s_h, \omega_h, a_h)$ over $\mathcal{B}$. Thus

$$\left\langle r_h^i(s_h, \omega_h, a_h, \cdot), \varphi_h^i \circ \nu_h(\cdot|s_h, \omega_h, a_h) - \nu_h(\cdot|s_h, \omega_h, a_h) \right\rangle_{\mathcal{B}} \leq 0, \forall \varphi_h^i \in \Phi^i.$$

which indicate for any policy $\pi$ and prior $\psi_h$, we have:

$$\left\langle r_h^i(s_h, \cdot, \cdot, \cdot), \psi_h(\cdot) \otimes \pi_h(\cdot|s_h) \otimes \left( \varphi_h^i \circ \nu_h(\cdot|s_h, \cdot, \cdot) - \nu_h(\cdot|s_h, \cdot, \cdot) \right) \right\rangle_{\Omega \times \mathcal{A} \times \mathcal{B}} \leq 0, \forall \varphi_h^i \in \Phi^i, i \times h \in [I] \times [H].$$

So there exists policy pair $\pi, \nu$ that satisfies BCE constraint. ∎

### B.2 CONSISTENCY OF PERFORMANCE METRIC

**Proposition B.2 (Consistency of Performance Metric)** $\mathrm{SubOpt}^{\pi,\nu}(s_1) = \mathrm{CV}^{\pi,\nu}(s_1) = 0$ *if and only if $\pi, \nu$ is optimal policy defined in Equation (2).*

**Proof** We can observe that $\mathrm{CV}^{\pi,\nu} = 0$ implies that for any $i \in [I]$, we also have $\mathrm{CV}^{i,\pi,\nu} = 0$. Therefore, for any state $s_h$ with a positive visitation measure with respect to $\pi$ and $\nu$, we have

$$\left\langle r_h^i(s_h), \psi_h \otimes \pi_h \otimes \left( \varphi_h^i \circ \nu_h - \nu_h \right) \right\rangle_{\Omega \times \mathcal{A} \times \mathcal{B}} (s_h) \leq 0, \forall \varphi_h^i \in \Phi^i.$$

We can note that for any $\tilde{\varphi}_h^i \in \Phi^i$, and any $\tilde{a} \in \mathcal{A}$ with $\pi_h(\tilde{a}|s_h) > 0$ in the definition, we could let $\varphi_h^i = \tilde{\varphi}_h^i$ if $a = \tilde{a}$, and $\varphi_h^i = I$ if $a \neq \tilde{a}$, where $I \circ \nu = \nu$ (no deviation). Thus, we can see that $\left\langle r_h^i, \psi_h \otimes \left( \tilde{\varphi}_h^i \circ \nu_h - \nu_h \right) \right\rangle_{\Omega \times \mathcal{B}} (s_h, \tilde{a}) \leq 0$. We conclude the proof by the arbitrariness of $\tilde{\varphi}_h^i$ and $\tilde{a}$. When combined with $\mathrm{SubOpt}^{\pi,\nu} = 0$, it implies that $\pi$ and $\nu$ constitute an optimal policy. The reverse direction is straightforward. ∎

## C PROOF OF THEOREM 4.1: LOWER BOUND

**Proof** Our proof borrows some idea from Cacciamani et al. (2023b). Consider the following instances, denoted X and Y. Instances X and Y are nearly identical, differing only in the rewards $\{r^i\}_{i \in [I]}$ and $R$. They share the following settings:

$$I = 2, |\mathcal{S}| = 1, \Omega = \{\omega, \omega'\}, |\mathcal{A}| = 1, \mathcal{B}_1 = \{a_1, a_2\}, \mathcal{B}_2 = \{b_1, b_2\}$$

where $|\cdot|$ denote the cardinality of a set. Since there are only two agents, we refer it to them as Agent-a and Agent-b, with their actions denoted as subscripts 1 and 2, respectively. The external parameter $\omega_1$ take two values $\omega$, and $\omega'$, with probabilities $\psi(\omega) = (1 + \varepsilon)/2, \psi(\omega') = (1 - \varepsilon)/2$. The agents' and principal's reward function is identical in different external parameters. When the action profile is $(a_i, b_j)$, where $i, j = 1, 2$, the reward is a random variable following a Bernoulli distribution with an expectation in the following Table.

Table 2: Pair of hard instances X (left) and Y (right)

| $r_1, r_2, R$ | $b_1$ | $b_2$ | | $r_1, r_2, R$ | $b_1$ | $b_2$ |
|---|---|---|---|---|---|---|
| $a_1$ | $1, \frac{1}{2}, 1$ | $0, \frac{1+\varepsilon}{2}, 0$ | | $a_1$ | $1, \frac{1}{2}, 1$ | $0, \frac{1-\varepsilon}{2}, 0$ |
| $a_2$ | $1, \frac{1}{2}, 1$ | $0, \frac{1+\varepsilon}{2}, 0$ | | $a_2$ | $1, \frac{1}{2}, 1$ | $0, \frac{1-\varepsilon}{2}, 0$ |

E.g. in instance X (left), when the action profile is $(a_1, b_2)$, no matter what external parameter is, Agent-a and principal will get 0, Agent-b will get $r_2 \sim \text{Ber}\left(\frac{1+\varepsilon}{2}\right)$. It is noted that the optimal actions in instance X are $(a_1, b_2)$ and $(a_2, b_2)$. In this case, agents do not have incentive to deviate from the recommended action, and the optimal value that principal can get is 0. However, in instance Y, the optimal actions are $(a_1, b_1)$ or $(a_2, b_1)$ and the principal's optimal reward is 1.

We denote the $t$-th policy $\nu^t(a_i, b_j|\omega_k)$ as $\nu_{ij}^{k,t}$, $i, j, k \in \{1, 2\}$. In $t$-th episode, we have the suboptimality in instance $Y$ is:

$$\text{SubOpt}_Y^t = 1 - \left( \frac{1+\varepsilon}{2}(\nu_{11}^{1,t} + \nu_{21}^{1,t}) + \frac{1-\varepsilon}{2}(\nu_{11}^{2,t} + \nu_{21}^{2,t}) \right),$$

where 1 is the principal's optimal reward, and the the term in bracket is the expected reward of following this policy. Meanwhile, the constraint violation in instance X for Agent-b (Agent-a's constraint violation is always 0) is:

$$\text{CV}_X^t = \left( \frac{(1+\varepsilon)}{2}(\nu_{11}^{1,t} \cdot \frac{\varepsilon}{2} + \nu_{21}^{1,t} \cdot \frac{\varepsilon}{2}) + \frac{(1-\varepsilon)}{2}(\nu_{11}^{2,t} \cdot \frac{\varepsilon}{2} + \nu_{21}^{2,t} \cdot \frac{\varepsilon}{2}) \right)$$

Then, we consider the regret of principal in instance Y after $T$ episodes:

$$R_Y^T = V_Y^{p,*} - V_Y^{p,T} = T - \sum_{t \in [T]} \left[ \frac{1+\varepsilon}{2}(\nu_{11}^{1,t} + \nu_{21}^{1,t}) + \frac{1-\varepsilon}{2}(\nu_{11}^{2,t} + \nu_{21}^{2,t}) \right]$$

$$= \sum_{t \in [T]} \left[ \frac{1+\varepsilon}{2}(\nu_{12}^{1,t} + \nu_{22}^{1,t}) + \frac{1-\varepsilon}{2}(\nu_{12}^{2,t} + \nu_{22}^{2,t}) \right].$$

For any algorithm $\mathscr{A}$, number $K$ there exists $\delta$ such that $\mathbb{P}_Y(R_Y^T \leq K) \geq 1 - \delta$ . Here, $\mathbb{P}_Y$ denotes the probability measure over the Canonical Bandit Model (Lattimore & Szepesvári, 2020) in instance Y, which is determined by algorithm and model. By Pinker's inequality:

$$\mathbb{P}_X(R_Y^T \leq K) \geq \mathbb{P}_Y(R_Y^T \leq K) - \sqrt{\frac{1}{2}\text{KL}(\mathbb{P}_Y||\mathbb{P}_X)}$$

$$\geq 1 - \delta - \sqrt{\frac{1}{2}\text{KL}(\mathbb{P}_Y||\mathbb{P}_X)}.$$

By divergence decomposition (Lattimore & Szepesvári, 2020) we have:

$$\text{KL}(\mathbb{P}_Y||\mathbb{P}_X) = \sum_{t \in [T]} \text{KL}\left( \text{Ber}\left(\frac{1+\varepsilon}{2}\right) || \text{Ber}\left(\frac{1-\varepsilon}{2}\right) \right)$$

$$\left[ \frac{1+\varepsilon}{2}\mathbb{E}_Y(\nu_{12}^{1,t}) + \frac{1-\varepsilon}{2}\mathbb{E}_Y(\nu_{12}^{2,t}) + \frac{1+\varepsilon}{2}\mathbb{E}_Y(\nu_{22}^{1,t}) + \frac{1-\varepsilon}{2}\mathbb{E}_Y(\nu_{22}^{2,t}) \right].$$

Note the definition of KL divergence:

$$\text{KL}\left( \text{Ber}\left(\frac{1+\varepsilon}{2}\right) || \text{Ber}\left(\frac{1-\varepsilon}{2}\right) \right) = \frac{1+\varepsilon}{2} \log \frac{1+\varepsilon}{1-\varepsilon} - \frac{1-\varepsilon}{2} \log \frac{1+\varepsilon}{1-\varepsilon}$$

$$= \varepsilon \log \frac{1+\varepsilon}{1-\varepsilon} \leq 4\varepsilon^2.$$

Then, by reverse Markov inequality, we have:

$$\mathbb{E}_Y \left[ \sum_{t \in [T]} \left( \frac{1+\varepsilon}{2}(\nu_{12}^{1,t} + \nu_{22}^{1,t}) + \frac{1-\varepsilon}{2}(\nu_{12}^{2,t} + \nu_{22}^{2,t}) \right) \right]$$

$$\leq (T - K)\mathbb{P}_Y(R_Y^T \geq K) + K$$

$$\leq (T - K)\delta + K.$$

Combine the above two inequalities, we have:

$$\mathbb{P}_X \left[ \sum_{t \in [T]} \left( \frac{1+\varepsilon}{2}(\nu_{12}^{1,t} + \nu_{22}^{1,t}) + \frac{1-\varepsilon}{2}(\nu_{12}^{2,t} + \nu_{22}^{2,t}) \right) \geq K \right] \geq 1 - \delta - \varepsilon\sqrt{2(T-K)\delta + 2K}.$$

Thus, we consider the constraint violation of instance X:

$$\text{CV}_X^T = \frac{\varepsilon}{4} \left[ \sum_{t \in [T]} \left( (1+\varepsilon)(\nu_{11}^{1,t} + \nu_{21}^{1,t}) + (1-\varepsilon)(\nu_{11}^{2,t} + \nu_{21}^{2,t}) \right) \right]$$

$$= \frac{\varepsilon}{4} \left[ 2T - \sum_{t \in [T]} \left( (1+\varepsilon)(\nu_{12}^{1,t} + \nu_{22}^{1,t}) + (1-\varepsilon)(\nu_{12}^{2,t} + \nu_{22}^{2,t}) \right) \right]$$

$$\geq \frac{1}{2}\varepsilon(T - K).$$

with probability at least $1 - \delta - \varepsilon\sqrt{2(T-K)\delta + 2K}$.

Let $K = \tilde{\mathcal{O}}(T^\alpha)$ and $\varepsilon = \tilde{\mathcal{O}}(T^{-\alpha/2})$. For any algorithm $\mathscr{A}$, if $R_Y^T = \tilde{\mathcal{O}}(T^\alpha)$ (otherwise the theorem is proved), then $\text{CV}_X^T = \Omega(T^{1-\alpha/2})$ with high probability. Thus, we have proved that for any algorithm, there exist $\delta$ such that with probability $1 - \delta$, either $\text{Reg}(T) = \tilde{\Omega}(T^\alpha)$ or $\text{CV}(T) = \tilde{\Omega}(T^{1-\alpha/2})$, i.e., we completed the proof. ∎

# D    PROOF OF REWARD-FREE GUARANTEE

In this section, we provide some theoretical guarantees for the reward-free stage as preparation for the proof of Theorem 4.4 in the next section.

**Lemma D.1 (Convergence rate of prior estimator)** *With probability at least $1 - \delta/6$, let $c_\omega = \sqrt{\ln(12H/\delta)}$, we have:*

$$\text{TV}(\psi_h^k, \psi_h) \leq c_\omega \sqrt{\frac{|\Omega|}{k}}, \forall k \in [K].$$

## D.1    PROOF OF LEMMA D.1

**Proof** For $k = 1$, it is trivial that:

$$\text{TV}(\psi_h^k, \psi_h) = \frac{1}{2}\|\psi_h^k - \psi_h\|_1 \leq \frac{1}{2} \sum_{\omega \in \Omega} |\psi_h^k(\omega) - \psi_h(\omega)| \leq 1.$$

When $k > 1$, according to (Qian et al., 2020), for any $\delta \in [0, 1]$, we have:

$$\mathbb{P}\left( \|\psi_h^{k+1} - \psi_h\|_1 \geq \sqrt{\frac{2|\Omega|\ln(2/\delta)}{k}} \right) \leq \delta.$$

We take union bound for $h \in [H]$, thus, with probability at least $1 - \delta/6$, we have:

$$\text{TV}(\psi_h^k, \psi_h) \leq \frac{1}{2}\sqrt{\frac{2|\Omega|\ln(12H/\delta)}{k-1}} \leq c_\omega\sqrt{\frac{|\Omega|}{k}}, \forall h \in [H]$$

where $c_\omega = \sqrt{\ln(12H/\delta)}$, and for any $k \in [K]$ the above inequality holds. ∎

To show the guarantee in the reward-free exploration, we define the $V$-function associated with the policy $\pi, \nu$ regarding the exploration-driven reward function $z^k$ in the $k$-th episode (line 6 in Appendix A.1) as $U_1^{\pi,\nu}(s_1; z^k)$. Moreover, let

$$U_1^*(s_1; z^k) = \max_{\pi,\nu} U_1^{\pi,\nu}(s_1; z^k).$$

where the maximum is taken over all policies $\pi$ and $\nu$ without any constraint. We also denote the maximizer as $(\pi^\dagger, \nu^\dagger)$. With this notation, we have the following guarantee:

**Lemma D.2 (Exploration Phase Guarantee)** *With probability at least $1 - \delta/2$, the exploration phase in Algorithm 1 satisfies for any $k \in [K]$*

$$U_1^*(s_1; z^k) \leq U_1^k(s_1) + \frac{2H^2 c_\omega |\Omega|^{\frac{1}{2}}}{\sqrt{k}}$$

*and*

$$\sum_{k=1}^K U_1^*(s_1^k; z^k) \leq c_1 \sqrt{H^4 d^3 |\Omega| K \log(dK/\delta)}.$$

*for some constant $c_1, c_\omega$, where $U_1^k$ is defined in Appendix A.1 (line 9).*

### D.2 PROOF OF LEMMA D.2

**Proof** In algorithm A.1, if we denote $\phi^\top \iota_h^k = P_h U_{h+1}^k$, the error of ridge regression in each step is given by:

$$\left| \phi^\top \iota_h^k - P_h U_{h+1}^k \right|$$

$$\leq \left| \phi^\top \left( \Lambda_h^k \right)^{-1} \sum_{\tau=1}^{k-1} \phi_h^\tau \left( U_{h+1}^k \left( s_{h+1}^\tau \right) - P_h U_{h+1}^k \left( s_h^\tau, \omega_h^\tau, a_h^\tau, b_h^\tau \right) \right) \right| + \left| \phi^\top \left( \Lambda_h^k \right)^{-1} \sum_{\tau=1}^{k-1} \phi_h^\tau \phi_h^\tau \iota_h^k - \phi^\top \iota_h^k \right|$$

$$\leq \|\phi\|_{\left( \Lambda_h^k \right)^{-1}} \left\| \sum_{\tau=1}^{k-1} \phi_h^\tau \left( U_{h+1}^k \left( s_{h+1}^\tau \right) - P_h U_{h+1}^k \left( s_h^\tau, \omega_h^\tau, a_h^\tau, b_h^\tau \right) \right) \right\|_{\left( \Lambda_h^k \right)^{-1}} + \lambda \|\phi\|_{\left( \Lambda_h^k \right)^{-1}} \|\iota_h^k\|_{\left( \Lambda_h^k \right)^{-1}}$$

and we note that:

$$\left\| \iota_h^k \right\|_{\left( \Lambda_h^k \right)^{-1}} = \left\| \left( \Lambda_h^k \right)^{-\frac{1}{2}} \iota_h^k \right\| \leq \left\| \left( \Lambda_h^k \right)^{-\frac{1}{2}} \right\| \cdot \left\| \xi_h + \int_S U_{h+1}^k(\cdot) d\mu_h(\cdot) \right\| \leq \frac{1}{\sqrt{\lambda}} \cdot 2H\sqrt{d}.$$

Then we restate Lemma B.3 in (Jin et al., 2020b):

**Lemma D.3** *For any $k \in [K]$ and $U_h^k$, $\Lambda_h^k$ defined in Sub-Algorithm A.1, we have:*

$$\left\| \sum_{\tau=1}^{k-1} \phi_h^\tau \left( U_{h+1}^k(s_{h+1}^\tau) - P_h U_{h+1}^k(s_h^\tau, \omega_h^\tau, a_h^\tau, b_h^\tau) \right) \right\|_{\left( \Lambda_h^k \right)^{-1}} \leq cdH\sqrt{\log(dK/\delta)}$$

*hold with probability at least $1 - \delta/6$ for some constant $c$*

Under the event in Lemma D.3 and recall the definition of $\beta = c_\beta dH\sqrt{\log(dK/\delta)}$ for some $c_\beta$, we have:

$$\left| \phi^\top \iota_h^k - P_h U_{h+1}^k \right| \leq \left( 2H\sqrt{\lambda d} + cdH\sqrt{\log(dK/\delta)} \right) \|\phi\|_{\left( \Lambda_h^k \right)^{-1}} \leq \beta \cdot \|\phi\|_{\left( \Lambda_h^k \right)^{-1}}.$$

If we define $\delta_h^k := z_h^k + P_h U_{h+1}^k - Q_h^k$, we can get:

$$\delta_h^k = z_h^k + P_h U_{h+1}^k - \left( \phi^\top \iota_h^k + z_h^k + u_h^k \right)$$

$$= -u_h^k + \left( P_h U_{h+1}^k - \phi^\top \iota_h^k \right).$$

Recall that $u_h^k = \beta \cdot \|\phi\|_{\left( \Lambda_h^k \right)^{-1}}$, then eventually we have:

$$-2\beta \cdot \|\phi\|_{\left( \Lambda_h^k \right)^{-1}} \leq \delta_h^k \leq 0.$$

Followed by the decomposition in (Wu et al., 2022), we have:

$$U_1^*(s_1, z^k) - U_1^k(s_1)$$

$$= \sum_{h \in [H]} \mathbb{E}^{\psi, \pi^\dagger, \nu^\dagger} \left[ \langle Q_h^k, \psi_h \otimes \pi_h^\dagger \otimes \nu_h^\dagger - \psi_h^k \otimes \pi_h^k \otimes \nu_h^k \rangle(s_h) \right] + \sum_{h \in [H]} \mathbb{E}^{\psi, \pi^\dagger, \nu^\dagger} \left[ \delta_h^k(s_h, \omega_h, a_h, b_h) \right]$$

$$= \sum_{h \in [H]} \mathbb{E}^{\psi, \pi^\dagger, \nu^\dagger} \left[ \langle Q_h^k, \psi_h^k \otimes \pi_h^\dagger \otimes \nu_h^\dagger - \psi_h^k \otimes \pi_h^k \otimes \nu_h^k \rangle(s_h) \right] + \sum_{h \in [H]} \mathbb{E}^{\psi, \pi^\dagger, \nu^\dagger} \left[ \delta_h^k(s_h, \omega_h, a_h, b_h) \right]$$

$$+ \sum_{h \in [H]} \mathbb{E}^{\psi, \pi^\dagger, \nu^\dagger} \left[ \langle Q_h^k, (\psi_h - \psi_h^k) \otimes \pi_h^\dagger \otimes \nu_h^\dagger \rangle(s_h) \right].$$

Since the $\pi_h^k, \nu_h^k$ is optimal policy with respect to $Q_h^k$, so the first term less than 0. Under the event in Lemma D.1, we can upper bound the third term by $\frac{2H^2 c_\omega |\Omega|^{\frac{1}{2}}}{\sqrt{k}}$. And under the event in Lemma D.3, we know that $\delta_h^k \leq 0$. Combine the above three inequalities, we have:

$$U_1^*(s_1, z^k) - U_1^k(s_1) \leq \frac{2H^2 c_\omega |\Omega|^{\frac{1}{2}}}{\sqrt{k}},$$

which completes the first part of the proof. For the second part, suppose we define:

$$\xi_h^k := P_h U_{h+1}^k \left(s_h^k, \omega_h^k, a_h^k, b_h^k\right) - U_{h+1}^k \left(s_{h+1}^k\right),$$

where $(s_h^k, \omega_h^k, a_h^k, b_h^k)$ is the trajectory generated by exploration phase in Algorithm 1 in $k$-th episode. Then we note that:

$$\begin{aligned}
U_h^k \left(s_h^k\right) &= z_h^k + u_h^k + \phi^\top \iota_h^k \\
&\leq \left(2 + \frac{1}{H}\right) \beta \cdot \|\phi_h^k\|_{(\Lambda_h^k)^{-1}} + P_h U_{h+1}^k \left(s_h^k, \omega_h^k, a_h^k, b_h^k\right) \\
&= U_{h+1}^k \left(s_{h+1}^k\right) + \xi_h^k + \left(2 + \frac{1}{H}\right) \beta \cdot \|\phi_h^k\|_{(\Lambda_h^k)^{-1}}.
\end{aligned}$$

Recursively use this formula, we have:

$$\begin{aligned}
U_1^k \left(s_1^k\right) &= \sum_{h=1}^{H} \left[U_h^k \left(s_h^k\right) - U_{h+1}^k \left(s_{h+1}^k\right)\right] \\
&= \sum_{h=1}^{H} \xi_h^k + \sum_{h=1}^{H} \left(2 + \frac{1}{H}\right) \beta \cdot \|\phi_h^k\|_{(\Lambda_h^k)^{-1}}.
\end{aligned}$$

So the sum:

$$\sum_{k=1}^{K} U_1^k \left(s_1^k\right) \leq \sum_{k=1}^{K} \sum_{h=1}^{H} \xi_h^k + \left(2 + \frac{1}{H}\right) \beta \sum_{k=1}^{K} \sum_{h=1}^{H} \|\phi_h^k\|_{(\Lambda_h^k)^{-1}}.$$

Define the filtration $\mathcal{F}_h^k$ as the $\sigma$-algebra generated by data in the exploration phase in Algorithm 1 up to and including step $h$ in the $k$-th episode. Then we note that $\mathbb{E}(\xi_h^k | \mathcal{F}_h^k) = 0$ and $|\xi_h^k| \leq 2H$, by Azuma-Hoeffding inequality, with probability at least $1 - \delta/6$ we have:

$$\sum_{k=1}^{K} \sum_{h=1}^{H} \xi_h^k \leq \sqrt{2KH^2 \log(6/\delta)}.$$

Then by elliptical potential lemma in (Abbasi-Yadkori et al., 2012), we have:

$$\sum_{k=1}^{K} \left(\phi_h^k\right)^\top \left(\Lambda_h^k\right)^{-1} \phi_h^k \leq 2 \log \left[\frac{\det \left(\Lambda_h^{K+1}\right)}{\det \left(\Lambda_h^1\right)}\right].$$

Moreover, note that $\left\|\Lambda_h^{K+1}\right\| = \left\|\sum_{k=1}^{K} \phi_h^k \left(\phi_h^k\right)^\top + \lambda I\right\| \leq \lambda + K$, by the Cauchy-Schwartz inequality, we have:

$$\sum_{k=1}^{K} \sum_{h=1}^{H} \sqrt{\left(\phi_h^k\right)^\top \left(\Lambda_h^k\right)^{-1} \phi_h^k} \leq \sum_{h=1}^{H} \sqrt{K} \cdot \left(\sum_{k=1}^{K} \left(\phi_h^k\right)^\top \left(\Lambda_h^k\right)^{-1} \phi_h^k\right)^{1/2} \leq H \sqrt{2dK \log \left(\frac{\lambda + K}{\lambda}\right)}.$$

Combine all the above inequalities, with probability at least $1 - \delta/2$:

$$\begin{aligned}
\sum_{k=1}^{K} U_1^*(s_1; z^k) &\leq \sum_{k=1}^{K} U_1^k(s_1) + \sum_{k=1}^{K} \frac{2H^2 c_\omega |\Omega|^{\frac{1}{2}}}{\sqrt{k}} \\
&\leq c \sqrt{H^4 d^3 K \log(dK/\delta)} + 4H^2 c_\omega |\Omega|^{\frac{1}{2}} \sqrt{K} \\
&\leq c_1 \sqrt{H^4 d^3 |\Omega| K \log(dK/\delta)}
\end{aligned}$$

for some constant $c_1$. Now we have completed the proof. ∎

# E  PROOF FOR THE THEOREM 4.4: EXPLORE-THEN-COMMIT BOUND

In this section, we first present the proof of Lemma 4.2. Next, we prove the formal version of Lemma 4.3, which provides the decomposition of suboptimality and constraint violation. Finally, we present the proof of our main theorem.

## E.1  PROOF OF LEMMA 4.2

**Proof** With a probability of at least $1 - \delta/3$ (given the event from Lemma D.1), for any policy pair $(\pi, \nu)$ satisfying the BCE Constraint 1 and for any $s \in \mathcal{S}$, we have:

$$
\left\langle \tilde{r}_h^i, \tilde{\psi}_h \otimes \pi_h \otimes \left( \hat{\varphi}_h^i \circ \nu_h - \nu_h \right) \right\rangle (s)
$$

$$
\leq \left\langle r_h^i + \tau_h, \tilde{\psi}_h \otimes \pi_h \otimes \hat{\varphi}_h^i \circ \nu_h \right\rangle (s) - \left\langle r_h^i, \tilde{\psi}_h \otimes \pi_h \otimes \nu_h \right\rangle (s)
$$

$$
\leq \left\langle \tau_h, \tilde{\psi}_h \otimes \pi_h \otimes \hat{\varphi}_h^i \circ \nu_h \right\rangle (s) + \left\langle r_h^i, \psi_h \otimes \pi_h \otimes \left( \hat{\varphi}_h^i \circ \nu_h - \nu_h \right) \right\rangle (s) + 4\,\mathrm{TV}(\psi_h, \tilde{\psi}_h)
$$

$$
= \left\langle \tau_h, \tilde{\psi}_h \otimes \pi_h \otimes \hat{\varphi}_h^i \circ \nu_h \right\rangle (s) + \frac{4c_\omega |\Omega|^{\frac{1}{2}}}{\sqrt{K}}.
$$

In the first inequality, we use the fact that $\left\langle r_h^i, \psi_h \otimes \pi_h \otimes \left( \hat{\varphi}_h^i \circ \nu_h - \nu_h \right) \right\rangle (s) = 0$ for all $s \in \mathcal{S}$ according to the definition of the BCE constraint. ∎

## E.2  FORMAL VERSION OF LEMMA 4.3

**Lemma E.1** *With probability at least $1 - \delta$, we have*

$$
\mathrm{SubOpt}^{\tilde{\pi}, \tilde{\nu}}(\tilde{s}_1) + \mathrm{CV}^{\tilde{\pi}, \tilde{\nu}}(\tilde{s}_1) \leq \sum_{h \in [H]} \mathrm{ReLU}\left( \mathbb{E}^{\pi^\star, \nu^\star}[\delta_h(s_h, \omega_h, a_h, b_h)] \right) + \mathrm{ReLU}\left( -\sum_{h \in [H]} \delta_h(\tilde{s}_h, \tilde{\omega}_h, \tilde{a}_h, \tilde{b}_h) \right)
$$

$$
+ \sum_{h \in [H]} \mathbb{E}^{\tilde{\pi}, \tilde{\nu}}\left[ \left\langle \tau_h, \psi_h \otimes \tilde{\pi}_h \otimes \tilde{\nu}_h \right\rangle (s_h) \right] + \sum_{h \in [H]} \mathbb{E}^{\tilde{\pi}, \tilde{\nu}}\left[ \left\langle \tau_h, \psi_h \otimes \tilde{\pi}_h \otimes \hat{\varphi}_h^i \circ \tilde{\nu}_h \right\rangle (s_h) \right] + \tilde{\mathcal{O}}\left( \frac{H^2 |\Omega|^{\frac{1}{2}}}{\sqrt{K}} \right),
$$

*where $\{\tilde{s}_h, \tilde{\omega}_h, \tilde{a}_h, \tilde{b}_h\}_{h \in [H]}$ is the trajectory generated by policy $(\tilde{\pi}, \tilde{\nu})$.*

**Proof** We first note the constraint violation in each step:

$$
\left\langle r_h^i, \psi_h \otimes \tilde{\pi}_h \otimes \hat{\varphi}_h^i \circ \tilde{\nu}_h \right\rangle (s_h) - \left\langle r_h^i, \psi_h \otimes \tilde{\pi}_h \otimes \tilde{\nu}_h \right\rangle (s_h)
$$

$$
\leq \left\langle \tilde{r}_h^i, \tilde{\psi}_h \otimes \tilde{\pi}_h \otimes \hat{\varphi}_h^i \circ \tilde{\nu}_h \right\rangle (s_h) - \left\langle \tilde{r}_h^i - \tau_h, \psi_h \otimes \tilde{\pi}_h \otimes \tilde{\nu}_h \right\rangle (s_h) + 2\,\mathrm{TV}(\psi_h, \tilde{\psi}_h)
$$

$$
\leq \left\langle \tau_h, \psi_h \otimes \tilde{\pi}_h \otimes \hat{\varphi}_h^i \circ \tilde{\nu}_h \right\rangle (s_h) + \left\langle \tau_h, \psi_h \otimes \tilde{\pi}_h \otimes \tilde{\nu}_h \right\rangle (s_h) + \frac{12c_\omega |\Omega|^{\frac{1}{2}}}{\sqrt{K}},
$$

where $\hat{\varphi}_h^i$ is the maximum constraint violation with respect to $(\tilde{\pi}, \tilde{\nu})$. Then by this result, it can be derived that:

$$
\mathrm{CV}^{i, \tilde{\pi}, \tilde{\nu}}(s_1) := \sum_{h=1}^{H} \mathbb{E}^{\tilde{\pi}, \tilde{\nu}}\left[ \left\langle r_h^i, \psi_h \otimes \tilde{\pi}_h \otimes \left( \hat{\varphi}_h^i \circ \tilde{\nu}_h - \tilde{\nu}_h \right) \right\rangle (s_h) \right]
$$

$$
\leq \mathbb{E}^{\tilde{\pi}, \tilde{\nu}}\left[ \sum_{h=1}^{H} \left\langle \tau_h, \psi_h \otimes \tilde{\pi}_h \otimes \tilde{\nu}_h \right\rangle (s_h) \right] + \mathbb{E}^{\tilde{\pi}, \tilde{\nu}}\left[ \sum_{h=1}^{H} \left\langle \tau_h, \psi_h \otimes \tilde{\pi}_h \otimes \hat{\varphi}_h^i \circ \tilde{\nu}_h \right\rangle (s_h) \right] + \frac{12c_\omega H |\Omega|^{\frac{1}{2}}}{\sqrt{K}}.
$$

Next, we focus on regret decomposition. Following Lemma 6.1 in (Wu et al., 2022), we have:

$$
V_1^{\pi^\star, \nu^\star}(s_1) - V_1^{\tilde{\pi}, \tilde{\nu}}(s_1) = \sum_{h \in [H]} \mathbb{E}^{\pi^\star, \nu^\star}\left[ \langle \tilde{Q}_h, \psi_h \otimes \pi_h^\star \otimes \nu_h^\star - \tilde{\psi}_h \otimes \tilde{\pi}_h \otimes \tilde{\nu}_h \rangle \right] + \sum_{h=1}^{H} \left( \zeta_h^1 + \zeta_h^2 \right)
$$

$$
+ \sum_{h \in [H]} \mathbb{E}^{\pi^\star, \nu^\star}[\delta_h(s_h, \omega_h, a_h, b_h)] + \sum_{h=1}^{H} \left\langle \tilde{Q}_h, (\tilde{\psi}_h - \psi_h) \otimes \tilde{\pi}_h \otimes \tilde{\nu}_h \right\rangle - \sum_{h=1}^{H} \delta_h(\tilde{s}_h, \tilde{\omega}_h, \tilde{a}_h, \tilde{b}_h),
$$

where $\delta_h = R_h + P_h \tilde{V}_{h+1} - \tilde{Q}_h$ and we denote $\{\tilde{s}_h, \tilde{\omega}_h, \tilde{a}_h, \tilde{b}_h\}_{h \in [H]}$ as a trajectory generated by policy $(\tilde{\pi}, \tilde{\nu})$ in Algorithm 1,

$$\zeta_h^1 = \left( \langle \tilde{Q}_h, \psi_h \otimes \tilde{\pi}_h \otimes \tilde{\nu}_h \rangle - V_h^{\tilde{\pi}, \tilde{\nu}} \right)(\tilde{s}_h) - (\tilde{Q}_h - Q_h^{\tilde{\pi}, \tilde{\nu}})(\tilde{s}_h, \tilde{\omega}_h, \tilde{a}_h, \tilde{b}_h)$$

$$\zeta_h^2 = P_h(\tilde{V}_h - V_{h+1}^{\tilde{\pi}, \tilde{\nu}})(\tilde{s}_h, \tilde{\omega}_h, \tilde{a}_h, \tilde{b}_h) - (\tilde{V}_{h+1} - V_h^{\tilde{\pi}, \tilde{\nu}})(\tilde{s}_{h+1}).$$

Similar to filtration defined in (Cai et al., 2020), we define $\mathcal{F}_{h,1}$ is the $\sigma$-algebra generated by $\{\tilde{s}_i, \tilde{\omega}_i, \tilde{a}_i, \tilde{b}_i\}_{i \in [h]}$ and $\mathcal{F}_{h,2}$ is the $\sigma$-algebra generated by: $\{\tilde{s}_i, \tilde{\omega}_i, \tilde{a}_i, \tilde{b}_i\}_{i \in [h]} \cup \{\tilde{s}_{h+1}\}$. Thus: $\zeta_1^1, \zeta_1^2, \cdots, \zeta_h^1, \zeta_h^2, \cdots, \zeta_H^1, \zeta_H^2$ is martingale difference sequence with respect to filtration: $\mathcal{F}_{1,1}, \mathcal{F}_{1,2}, \cdots, \mathcal{F}_{h,1}, \mathcal{F}_{h,2}, \cdots, \mathcal{F}_{H,1}, \mathcal{F}_{H,2}$ since it is easy to verify that $\zeta_h^j \in \mathcal{F}_{h,j}$, for $j = 1, 2$ and $\mathbb{E}\left[ \zeta_h^1 | \mathcal{F}_{h-1,2} \right] = \mathbb{E}\left[ \zeta_h^2 | \mathcal{F}_{h,1} \right] = 0$. Then by the Azuma-Hoeffding inequality:

$$\sum_{h=1}^{H} (\zeta_h^1 + \zeta_h^2) \le \sqrt{8H^3 \log(6/\delta)}$$

hold with probability at least $1 - \delta/6$. Note that, according to Lemma 4.2 and Lemma D.1,

$$\sum_{h=1}^{H} \langle \tilde{Q}_h, \psi_h \otimes \pi_h^\star \otimes \nu_h^\star - \tilde{\psi}_h \otimes \tilde{\pi}_h \otimes \tilde{\nu}_h \rangle$$

$$= \sum_{h=1}^{H} \langle \tilde{Q}_h, \tilde{\psi}_h \otimes \pi_h^\star \otimes \nu_h^\star - \tilde{\psi}_h \otimes \tilde{\pi}_h \otimes \tilde{\nu}_h \rangle + \sum_{h=1}^{H} \langle \tilde{Q}_h, (\tilde{\psi}_h - \psi_h) \otimes \pi_h^\star \otimes \nu_h^\star \rangle$$

$$\le \frac{2H^2 c_\omega |\Omega|^{\frac{1}{2}}}{\sqrt{K}}.$$

And by Lemma D.1, we have:

$$\sum_{h=1}^{H} \langle \tilde{Q}_h, (\tilde{\psi}_h - \psi_h) \otimes \tilde{\pi}_h \otimes \tilde{\nu}_h \rangle \le \frac{2H^2 c_\omega |\Omega|^{\frac{1}{2}}}{\sqrt{K}}.$$

Note that for any $a, b \in \mathbb{R}$, $\mathrm{ReLU}(a + b) \le \mathrm{ReLU}(a) + \mathrm{ReLU}(b)$, we completed the proof. ∎

### E.3 Proof of Theorem 4.4

**Proof** Leveraging Lemma E.1, we need to provide a bound on $\delta_h$ and show the specific form of $\tau_h$. Firstly, we show that $\tau_h = 2\gamma \cdot \|\phi\|_{\Lambda_h^{-1}}$ is an uncertainty function, i.e., it satisfies $0 \le \tilde{r}_h^i - r_h^i \le 2\gamma \cdot \|\phi\|_{\Lambda_h^{-1}}$. We recall that we get $\tilde{\theta}_h^i$ by ridge regression on dataset $\mathcal{D}$, thus:

$$\left| \phi^\top \tilde{\theta}_h^i - \phi^\top \theta_h^i \right|$$

$$= \left| \phi^\top \Lambda_h^{-1} \sum_{\tau=1}^{k} \phi_h^\tau \cdot r_h^\tau - \phi^\top \theta_h^i \right|$$

$$\le \left| \phi^\top \Lambda_h^{-1} \sum_{\tau=1}^{k} \phi_h^\tau \left( r_h^\tau - (\phi_h^\tau)^\top \theta_h^i \right) \right| + \left| \phi^\top \Lambda_h^{-1} \sum_{\tau=1}^{k} \phi_h^\tau (\phi_h^\tau)^\top \theta_h^i - \phi^\top \theta_h^i \right|$$

$$\le \|\phi\|_{\Lambda_h^{-1}} \left( \left\| \sum_{\tau=1}^{k} \phi_h^\tau \left( r_h^\tau - (\phi_h^\tau)^\top \theta_h^i \right) \right\|_{\Lambda_h^{-1}} + \|\theta_h^i\|_{\Lambda_h^{-1}} \right).$$

Note that:

$$\|\theta_h^i\|_{\Lambda_h^{-1}} \le \left\| \Lambda_h^{-\frac{1}{2}} \theta_h^i \right\| \le \left\| \Lambda_h^{-\frac{1}{2}} \right\| \cdot \|\theta_h^i\| \le \frac{1}{\sqrt{\lambda}} \sqrt{d}$$

and $\left| r_h^\tau - (\phi_h^\tau)^\top \theta_h^i \right| \leq 2$, according to (Abbasi-Yadkori et al., 2012), with probability at least $1 - \delta/4$:

$$\left\| \sum_{\tau=1}^K \phi_h^\tau \left( r_h^\tau - (\phi_h^\tau)^\top \theta_h^i \right) \right\|_{\Lambda_h^{-1}} \leq 2\sqrt{\log\left( \frac{36 \det |\Lambda_h|}{\det |\lambda I| \delta^2} \right)}$$

$$\leq 2\sqrt{d \cdot \log\left( \frac{36(\lambda + K)}{\delta^2 \lambda} \right)}.$$

Thus, with the probability at least $1 - \delta/6$:

$$\left| \phi^\top \tilde{\theta}_h^i - \phi^\top \theta_h^i \right| \leq c_\gamma \sqrt{d \log(K/\delta)} \cdot \|\phi\|_{\Lambda_h^{-1}}$$

for some constant $c_\gamma$. Recall the definition of $\gamma = c_\gamma \sqrt{d \log(K/\delta)}$, we have:

$$0 \leq \tilde{r}_h^i - r_h^i = \phi^\top \tilde{\theta}_h^i - \phi^\top \theta_h^i + v_h \leq 2\gamma \cdot \|\phi\|_{\Lambda_h^{-1}}.$$

Then following the proof in Lemma D.2, we also have:

$$-2\beta \cdot \|\phi\|_{\Lambda_h^{-1}} \leq \delta_h \leq 0.$$

Now, let us tackle with the term in Lemma E.1. It is easy to see $\Lambda_h^k \preceq \Lambda_h$, thus $\|\cdot\|_{(\Lambda_h^k)^{-1}} \geq \|\cdot\|_{\Lambda_h^{-1}}$. Using this property, we have

$$\mathbb{E}^{\tilde{\pi}, \tilde{\nu}} \left[ \sum_{h=1}^H \left\langle 2\gamma \cdot \|\phi\|_{\Lambda_h^{-1}}, \psi_h \otimes \tilde{\pi}_h \otimes \tilde{\nu}_h \right\rangle (s_h) \right]$$

$$= \frac{1}{K} \sum_{k=1}^K \mathbb{E}^{\tilde{\pi}, \tilde{\nu}} \left[ \sum_{h=1}^H \left\langle 2\gamma \cdot \|\phi\|_{\Lambda_h^{-1}}, \psi_h \otimes \tilde{\pi}_h \otimes \tilde{\nu}_h \right\rangle (s_h) \right]$$

$$\leq \frac{1}{K} \sum_{k=1}^K \mathbb{E}^{\tilde{\pi}, \tilde{\nu}} \left[ \sum_{h=1}^H \left\langle 2\gamma \cdot \|\phi\|_{(\Lambda_h^k)^{-1}}, \psi_h \otimes \tilde{\pi}_h \otimes \tilde{\nu}_h \right\rangle (s_h) \right].$$

Given our consideration of $\frac{\beta}{H} \|\phi\|_{(\Lambda_h^k)^{-1}}$ as the reward function, along with Lemma D.2, with a probability of at least $1 - \delta/2$, we have

$$\frac{1}{K} \sum_{k=1}^K \mathbb{E}^{\tilde{\pi}, \tilde{\nu}} \left[ \sum_{h=1}^H \left\langle 2\gamma \cdot \|\phi\|_{(\Lambda_h^k)^{-1}}, \psi_h \otimes \tilde{\pi}_h \otimes \tilde{\nu}_h \right\rangle (s_h) \right]$$

$$= \frac{1}{K} \frac{2\gamma H}{\beta} \sum_{k=1}^K U_1^{\tilde{\pi}, \tilde{\nu}} \left( s_1; z^k \right)$$

$$\leq \frac{2\gamma H}{\beta} \frac{1}{K} \sum_{k=1}^K U_1^* \left( s_1; z^k \right)$$

$$\leq \frac{2c_\gamma c_1}{c_\beta} \sqrt{H^4 d^2 |\Omega| \log(K/\delta)} \cdot \frac{1}{\sqrt{K}}.$$

Then note that the visitation measure $\bar{d}_h^{\tilde{\pi}, \hat{\varphi}_h^i \circ \tilde{\nu}}$ generated by policy $(\tilde{\pi}_1, \tilde{\nu}_1, \cdots, \tilde{\pi}_{h-1}, \tilde{\nu}_{h-1}, \tilde{\pi}_h, \hat{\varphi}_h^i \circ \tilde{\nu}_h, \cdots, \tilde{\pi}_H, \tilde{\nu}_H)$ have the property that: $\bar{d}_h^{\tilde{\pi}, \tilde{\nu}}(s_h) = \bar{d}_h^{\tilde{\pi}, \hat{\varphi}_h^i \circ \tilde{\nu}}(s_h), \forall s_h \in \mathcal{S}, h \in [H]$. Thus for any

$h \in [H]$, we have:

$$\mathbb{E}^{\tilde{\pi}, \tilde{\nu}} \left[ \left\langle 2\gamma \cdot \|\phi\|_{\Lambda_h^{-1}}, \psi_h \otimes \tilde{\pi}_h \otimes \hat{\varphi}_h^i \circ \tilde{\nu}_h \right\rangle (s_h) \right]$$

$$\leq \frac{1}{K} \frac{2\gamma H}{\beta} \sum_{k=1}^{K} \mathbb{E}^{\tilde{\pi}, \tilde{\nu}} \left[ \left\langle \frac{\beta}{H} \|\phi\|_{(\Lambda_h^k)^{-1}}, \psi_h \otimes \tilde{\pi}_h \otimes \hat{\varphi}_h^i \circ \tilde{\nu}_h \right\rangle (s_h) \right]$$

$$= \frac{1}{K} \frac{2\gamma H}{\beta} \sum_{k=1}^{K} \left[ \sum_{s_h \in \mathcal{S}} \bar{d}_h^{\tilde{\pi}, \hat{\varphi}^i \circ \tilde{\nu}}(s_h) \left\langle \frac{\beta}{H} \|\phi\|_{(\Lambda_h^k)^{-1}}, \psi_h \otimes \tilde{\pi}_h \otimes \hat{\varphi}_h^i \circ \nu_h \right\rangle (s_h) \right]$$

$$\leq \frac{1}{K} \frac{2\gamma H}{\beta} \sum_{k=1}^{K} U_1^{\tilde{\pi}, \hat{\varphi}_h^i \circ \tilde{\nu}}(s_1; z^k).$$

In the last inequality, we magnify the expected error $\mathbb{E}^{\tilde{\pi}, \tilde{\nu}} \left[ \left\langle 2\gamma \cdot |\phi|_{(\Lambda_h^k)^{-1}}, \psi_h \otimes \tilde{\pi}_h \otimes \hat{\varphi} h^i \circ \tilde{\nu} h \right\rangle (s_h) \right]$, to the expectation $U_1^{\tilde{\pi}, \hat{\varphi}_h^i \circ \tilde{\nu}}(s_1; z^k)$, which is the summation of all $H$ steps. Then, we have:

$$\mathbb{E}^{\tilde{\pi}, \tilde{\nu}} \left[ \sum_{h=1}^{H} \left\langle 2\gamma \cdot \|\phi\|_{\Lambda_h^{-1}}, \psi_h \otimes \tilde{\pi}_h \otimes \hat{\varphi}_h^i \circ \tilde{\nu}_h \right\rangle (s_h) \right] \leq \frac{c_\gamma c_1}{2c_\beta} \sqrt{H^6 d^2 |\Omega| \log(K/\delta)} \cdot \frac{1}{\sqrt{K}}.$$

We note that according to Lemma D.2, with probability at least $1 - \delta/2$:

$$-\sum_{h=1}^{H} \delta_h(\tilde{s}_h, \tilde{\omega}_h, \tilde{a}_h, \tilde{b}_h) \leq \sum_{h=1}^{H} 2\beta \|\phi(\tilde{s}_h, \tilde{\omega}_h, \tilde{a}_h, \tilde{b}_h)\|_{\Lambda_h^{-1}}$$

$$\leq \frac{2H}{K} \sum_{k=1}^{K} \sum_{h=1}^{H} \frac{\beta}{H} \|\phi(\tilde{s}_h, \tilde{\omega}_h, \tilde{a}_h, \tilde{b}_h)\|_{(\Lambda_h^k)^{-1}}$$

$$\leq \frac{2H}{K} \sum_{k=1}^{K} U_1^{\star}(\tilde{s}_1; z^k)$$

$$\leq 2c_1 \sqrt{H^6 d^3 |\Omega| \log(dK/\delta)} \cdot \frac{1}{\sqrt{K}}.$$

Thus, we have proved that with probability at least $1 - \delta$, we have:

$$\text{SubOpt}^{\tilde{\pi}, \tilde{\nu}}(s_1) + \max_{i \in [I]} \text{CV}^{i, \tilde{\pi}, \tilde{\nu}}(s_1) \leq c_2' \sqrt{H^6 d^3 |\Omega| \log(dK/\delta)} \cdot \frac{1}{\sqrt{K}}$$

for some constant $c_2'$. Thus the regret and constraint violation in Algorithm 1 is:

$$\sum_{t \in [T]} \left[ \text{SubOpt}^{\tilde{\pi}, \tilde{\nu}}(s_1^t) + \text{CV}^{\pi^t, \nu^t}(s_1^t) \right]$$

$$\leq \sum_{t=1}^{K} \left[ \text{SubOpt}^{\tilde{\pi}, \tilde{\nu}}(s_1^t) + \text{CV}^{\pi^t, \nu^t}(s_1^t) \right] + \sum_{t=K+1}^{T} \left[ \text{SubOpt}^{\tilde{\pi}, \tilde{\nu}}(s_1^t) + \text{CV}^{\pi^t, \nu^t}(s_1^t) \right]$$

$$\leq K \cdot 4H + (T - K) \cdot c_2' \sqrt{H^6 d^3 |\Omega| \log(dK/\delta)} \frac{1}{\sqrt{K}}.$$

Let $K = T^{\frac{2}{3}}$. Then, with probability at least $1 - \delta$, we have:

$$\sum_{t \in [T]} \left[ \text{SubOpt}^{\pi^t, \nu^t}(s_1^t) + \text{CV}^{\pi^t, \nu^t}(s_1^t) \right] \leq c_2 \sqrt{H^6 d^3 |\Omega| \log(dT/\delta)} \cdot T^{\frac{2}{3}}$$

for some constant $c_2$. Equivalently, we have proved:

$$\sum_{t \in [T]} \left[ \text{SubOpt}^{\pi^t, \nu^t}(s_1^t) + \text{CV}^{\pi^t, \nu^t}(s_1^t) \right] = \tilde{\mathcal{O}} \left( \sqrt{H^6 d^3 |\Omega|} \cdot T^{\frac{2}{3}} \right)$$

with probability at least $1 - \delta$. ∎

## F   PROOF OF THEOREM 5.1: ADDITIONAL FEEDBACK BOUND

**Proof**   Firstly, we show that $\tau_h^t = 2\gamma \cdot \|\phi\|_{(\Lambda_h^t + \Lambda_h^{i,t})^{-1}}$ is an uncertainty function. Similar to the proof of Lemma 4.4, we first tackle with the term in ridge regression. And we use the following notation to describe the $\tau$-th episode data in step $h$:

$$\begin{cases} \phi_h^\tau = \phi(s_h^\tau, \omega_h^\tau, a_h^\tau, b_h^\tau) \\ r_h^{i,\tau} = r_h^i\left((s_h^\tau, \omega_h^\tau, a_h^\tau, b_h^\tau)\right) \\ \hat{\phi}_h^{i,\tau} = \hat{\phi}(s_h^\tau, \omega_h^\tau, a_h^\tau, \hat{b}_h^{i,\tau}, b_h^{-i,\tau}) \\ \hat{r}_h^{i,\tau} = r_h^i\left((s_h^\tau, \omega_h^\tau, a_h^\tau, \hat{b}_h^{i,\tau}, b_h^{-i,\tau})\right) \end{cases}$$

and the other variables are defined in Algorithm 2. Then we have:

$$\left| \phi^\top \theta_h^{i,t} - \phi^\top \theta_h^i \right|$$

$$= \left| \phi^\top (\Lambda_h^t + \Lambda_h^{i,t})^{-1} \sum_{\tau=1}^{t-1} (\phi_h^\tau \cdot r_h^{i,\tau} + \hat{\phi}_h^{i,\tau} \cdot \hat{r}_h^{i,\tau}) - \phi^\top \theta_h^i \right|$$

$$\leq \underbrace{\left| \phi^\top (\Lambda_h^t + \Lambda_h^{i,t})^{-1} \left( \sum_{\tau=1}^{t-1} \phi_h^\tau \cdot (r_h^{i,\tau} - \phi_h^\tau \theta_h^i) + \sum_{\tau=1}^{t-1} \hat{\phi}_h^{i,\tau} \cdot (\hat{r}_h^{i,\tau} - \hat{\phi}_h^{i,\tau} \theta_h^i) \right) \right|}_{\text{Term i}} +$$

$$\underbrace{\left| \phi^\top (\Lambda_h^t + \Lambda_h^{i,t})^{-1} \left( \sum_{\tau=1}^{t-1} \phi_h^\tau (\phi_h^\tau)^\top + \hat{\phi}_h^{i,\tau} (\hat{\phi}_h^{i,\tau})^\top \right) \theta_h^i - \phi^\top \theta_h^i \right|}_{\text{Term ii}}.$$

For the first term, we have:

$$\left| \phi^\top (\Lambda_h^t + \Lambda_h^{i,t})^{-1} \left( \sum_{\tau=1}^{t-1} \phi_h^\tau \cdot (r_h^{i,\tau} - \phi_h^\tau \theta_h^i) + \sum_{\tau=1}^{t-1} \hat{\phi}_h^{i,\tau} \cdot (\hat{r}_h^{i,\tau} - \hat{\phi}_h^{i,\tau} \theta_h^i) \right) \right|$$

$$\leq \|\phi\|_{(\Lambda_h^t + \Lambda_h^{i,t})^{-1}} \left\| \sum_{\tau=1}^{t-1} \phi_h^\tau \cdot (r_h^{i,\tau} - \phi_h^\tau \theta_h^i) + \sum_{\tau=1}^{t-1} \hat{\phi}_h^{i,\tau} \cdot (\hat{r}_h^{i,\tau} - \hat{\phi}_h^{i,\tau} \theta_h^i) \right\|_{(\Lambda_h^t + \Lambda_h^{i,t})^{-1}}$$

$$\leq \|\phi\|_{(\Lambda_h^t + \Lambda_h^{i,t})^{-1}} \left( \left\| \sum_{\tau=1}^{t-1} \phi_h^\tau \cdot (r_h^{i,\tau} - \phi_h^\tau \theta_h^i) \right\|_{(\Lambda_h^t + \Lambda_h^{i,t})^{-1}} + \left\| \sum_{\tau=1}^{t-1} \hat{\phi}_h^{i,\tau} \cdot (\hat{r}_h^{i,\tau} - \hat{\phi}_h^{i,\tau} \theta_h^i) \right\|_{(\Lambda_h^t + \Lambda_h^{i,t})^{-1}} \right)$$

$$\leq \|\phi\|_{(\Lambda_h^t + \Lambda_h^{i,t})^{-1}} \left( \left\| \sum_{\tau=1}^{t-1} \phi_h^\tau \cdot (r_h^{i,\tau} - \phi_h^\tau \theta_h^i) \right\|_{(\Lambda_h^t)^{-1}} + \left\| \sum_{\tau=1}^{t-1} \hat{\phi}_h^{i,\tau} \cdot (\hat{r}_h^{i,\tau} - \hat{\phi}_h^{i,\tau} \theta_h^i) \right\|_{(\Lambda_h^{i,t})^{-1}} \right),$$

where the first inequality is due to the Cauchy-Schwarz inequality and the second inequality arises from the triangle inequality in the norm $\| \cdot \|_{(\Lambda_h^t + \Lambda_h^{i,t})^{-1}}$. Then, the third inequality stems from $\Lambda_h^t \preceq \Lambda_h^t + \Lambda_h^{i,t}$ and $\Lambda_h^{i,t} \preceq \Lambda_h^t + \Lambda_h^{i,t}$, which implies:

$$\| \cdot \|_{(\Lambda_h^t + \Lambda_h^{i,t})^{-1}} \leq \| \cdot \|_{(\Lambda_h^{i,t})^{-1}} \text{ and } \| \cdot \|_{(\Lambda_h^t + \Lambda_h^{i,t})^{-1}} \leq \| \cdot \|_{(\Lambda_h^t)^{-1}}.$$

Then, through the self-normalization process in (Abbasi-Yadkori et al., 2012) and the auxiliary lemma in (Wu et al., 2022), we obtain $\det(\Lambda_h^t) \leq (\lambda + \frac{T}{d})^d$ and $\det(\Lambda_h^{i,t}) \leq (\lambda + \frac{T}{d})^d$ for all $i \in [I]$. Thus, with probability at least $1 - \delta/6$, we have:

$$\left\| \sum_{\tau=1}^{t-1} \phi_h^\tau \cdot (r_h^{i,\tau} - \phi_h^\tau \theta_h^i) \right\|_{(\Lambda_h^t)^{-1}} + \left\| \sum_{\tau=1}^{t-1} \hat{\phi}_h^{i,\tau} \cdot (\hat{r}_h^{i,\tau} - \hat{\phi}_h^{i,\tau} \theta_h^i) \right\|_{(\Lambda_h^{i,t})^{-1}} \leq 4\sqrt{d \log\left(\frac{T + \lambda d}{\lambda d}\right) + 2\log(\frac{6}{\delta})}.$$

And for the second term:

$$\left| \phi^\top (\Lambda_h^t + \Lambda_h^{i,t})^{-1} \left( \sum_{\tau=1}^{t-1} \phi_h^\tau (\phi_h^\tau)^\top + \hat{\phi}_h^{i,\tau} (\hat{\phi}_h^{i,\tau})^\top \right) \theta_h^i - \phi^\top \theta_h^i \right|$$

$$\leq \|\phi\|_{(\Lambda_h^t + \Lambda_h^{i,t})^{-1}} 2\lambda \|\theta_h^i\|_{(\Lambda_h^t + \Lambda_h^{i,t})^{-1}}$$

$$\leq \|\phi\|_{(\Lambda_h^t + \Lambda_h^{i,t})^{-1}} 2\lambda \|\theta_h^i\|_{(2\lambda I)^{-1}}$$

$$\leq \|\phi\|_{(\Lambda_h^t + \Lambda_h^{i,t})^{-1}} \sqrt{2\lambda d}.$$

In conclusion, we have proved that:

$$\left| \phi^\top \theta_h^{i,t} - \phi^\top \theta_h^i \right| \leq c_\gamma \sqrt{d \log\left( \frac{T + \lambda d}{\lambda d \delta} \right)} \|\phi\|_{(\Lambda_h^t + \Lambda_h^{i,t})^{-1}} \leq \gamma \|\phi\|_{(\Lambda_h^t + \Lambda_h^{i,t})^{-1}}$$

for some constant $c_\gamma$ and $\gamma = c_\gamma \sqrt{d \log(T/\delta)}$, which also suggests that:

$$0 \leq r_h^{i,t} - r_h^i = \phi^\top \theta_h^{i,t} + v_h^t - \phi^\top \theta_h^i \leq 2\gamma \cdot \|\phi\|_{(\Lambda_h^t + \Lambda_h^{i,t})^{-1}}.$$

Then, applying Lemma E.1, we first show that the constraint violation in $t$-th episode, with probability at least $1 - \delta/3$, we have:

$$\mathbb{E}^{\pi^t, \nu^t} \left[ \sum_{h=1}^H \left\langle 2\gamma \|\phi\|_{(\Lambda_h + \Lambda_h^{i,t})^{-1}}, \psi_h \otimes \pi_h^t \otimes (\nu_h^t + \hat{\varphi}_h^{i,t} \circ \nu_h^t) \right\rangle (s_h) \right]$$

$$= 2\gamma \mathbb{E} \left[ \sum_{h=1}^H \|\phi(s_h^t, \omega_h^t, a_h^t, b_h^t)\|_{(\Lambda_h^t + \Lambda_h^{i,t})^{-1}} \right] + 2\gamma \mathbb{E} \left[ \sum_{h=1}^H \|\phi(s_h^t, \omega_h^t, a_h^t, \hat{b}_h^{i,t}, b_h^{-i,t})\|_{(\Lambda_h^t + \Lambda_h^{i,t})^{-1}} \right]$$

$$\leq 2\gamma \mathbb{E} \left[ \sum_{h=1}^H \|\phi(s_h^t, \omega_h^t, a_h^t, b_h^t)\|_{(\Lambda_h^t)^{-1}} \right] + 2\gamma \mathbb{E} \left[ \sum_{h=1}^H \|\phi(s_h^t, \omega_h^t, a_h^t, \hat{b}_h^{i,t}, b_h^{-i,t})\|_{(\Lambda_h^{i,t})^{-1}} \right],$$

where $\{s_h^t, \omega_h^t, a_h^t, b_h^t\}_{t \times h \in [T] \times [H]}$ is the trajectory generated by Algorithm 2. We have:

$$2\gamma \sum_{t=1}^T \sum_{h=1}^H \|\phi(s_h^t, \omega_h^t, a_h^t, b_h^t)\|_{(\Lambda_h^t)^{-1}} \leq 2\gamma \sum_{h=1}^H \sqrt{T} \sqrt{2 \log\left( \frac{\det(\Lambda_h^T)}{\det(\lambda I)} \right)}$$

$$\leq 2\gamma H \sqrt{T} \sqrt{2d \log\left( \frac{\lambda d + T}{\lambda d} \right)}.$$

Since Cauchy-Schwartz inequality and Elliptical potential in (Abbasi-Yadkori et al., 2012). Similarly, we also have:

$$2\gamma \sum_{t=1}^T \sum_{h=1}^H \|\phi(s_h^t, \omega_h^t, a_h^t, \hat{b}_h^{i,t}, b_h^{-i,t})\|_{(\Lambda_h^{i,t})^{-1}} \leq 2\gamma H \sqrt{T} \sqrt{2d \log\left( \frac{\lambda d + T}{\lambda d} \right)},$$

where $\{s_h^t, \omega_h^t, a_h^t, \hat{b}_h^{i,t}, b_h^{-i,t}\}_{t \times h \in [T] \times [H]}$ is the trajectory generated by algorithm 2 as well and $\hat{b}_h^{i,t}$ is the additional feedback reported by $i$-th agent in episode $t$. And for the last term, we have:

$$\sum_{t=1}^T \frac{8 H c_\omega |\Omega|}{\sqrt{t}} \leq 16 H c_\omega \sqrt{T}.$$

Then the proof is identical to the proof in Lemma D.2, we also have:

$$-2\beta \cdot \|\phi\|_{(\Lambda_h^t)^{-1}} \leq \delta_h^t \leq 0, \forall t \in [T].$$

And we note that:

$$-\sum_{t=1}^T \sum_{h=1}^H \delta_h^t (s_h^t, \omega_h^t, a_h^t, b_h^t) \leq \sum_{t=1}^T \sum_{h=1}^H 2\beta \|\phi(s_h^t, \omega_h^t, a_h^t, b_h^t)\|_{(\Lambda_h^t)^{-1}}$$

$$\leq 2\beta H \sqrt{T} \sqrt{2d \log\left( \frac{\lambda d + T}{\lambda d} \right)}.$$

Thus, we have proved that with probability at least $1 - 2\delta/3$:

$$\sum_{t=1}^{T} \frac{4H^2 c_\omega |\Omega|}{\sqrt{t}} + 2\beta H \sqrt{T} \sqrt{2d \log\left(\frac{\lambda d + T}{\lambda d}\right)} \leq 8H^2 c_\omega |\Omega| \sqrt{t} + 2\beta H \sqrt{T} \sqrt{2d \log\left(\frac{\lambda d + T}{\lambda d}\right)}$$

$$\leq C_2' \log(dK/\delta) \sqrt{H^4 d^3 |\Omega| T}$$

for some constant $C_2'$. Thus, we have:

$$\sum_{t \in [T]} \left[ \text{SubOpt}^{\pi^t, \nu^t}(s_1^t) + \text{CV}^{\pi^t, \nu^t}(s_1^t) \right] = \tilde{\mathcal{O}}\left( \sqrt{H^4 d^3 |\Omega|} \cdot \sqrt{T} \right)$$

with probability at least $1 - \delta$. ∎