# OpenReview forum: "Steer a Crowd: Learning to Persuade a Population in a Stackelberg Game"
_ICLR.cc/2025/Conference — ICLR 2025 Conference Withdrawn Submission_

### Official Review · Reviewer_vxG2 · 2024-10-27

**Soundness:** 3
**Presentation:** 2
**Contribution:** 2
**Rating:** 5
**Confidence:** 3

**Summary:**

This work studies information design in Markov games. Precisely, the authors study online settings in which the learner (i.e., the principal) aims to maximize her rewards (that is, minimizing the regret) while satisfying incentive compatibility constraints (in this paper, measured by the constraint violations performance metric). Since the environment is completely unknown to the learner, the agents are assumed to strictly follows the principal recommendations. The paper provides a lower-bound on the regret and violations the learner may attain when standard bandit feedback is available. An algorithm tailored for this specific setting is proposed; the algorithm guarantees $T^{2/3}$ regret and violations. When additional feedback is available, the authors propose an algorithm which attains $\sqrt{T}$ regret and violations.

**Strengths:**

This is the first work to study online information design in Markov games when no information about the environment is available.

**Weaknesses:**

To me, the main weakness of this work is the lack of sufficient theoretical contribution for the community. Specifically, the authors do not mention the work that is mainly related to theirs. Precisely, [Bacchiocchi et al. 2024] studies a similar setting to the one presented in this work. The main difference is that [Bacchiocchi et al. 2024]  focuses on tabular MDPs for information design, while this work generalize some of their results for Markov games.
First of all, notice that, the lower bound presents a similar result. Morover, the resolution techniques for both the easy setting and the bandit one are based on UCB for the latter, and EC for the first. On the contrary, (differently from [Bacchiocchi et al. 2024]) the upper bound on regret and violations presented in the work are clearly not tight, while only the optimal trade-off between regret and violations is attained.
Overall, I do not believe that an extension from episodic tabular MDPs to general Markov games is sufficient to meet the acceptance bar, since, the techniques employed to deal with the information design can be easily taken from [Bernasconi et al. 2022], [Cacciamani et al. 2023] and [Bacchiocchi et al. 2024], while the techniques employed for Markov games can be taken from many related works in RL such as [Azar et al. 2017] or [Jin et al. 2020].

Minor:
 I would suggest to split the result of regret and violations and bounding directly the sum of the performance metrics.

[Bacchiocchi et al. 2024] "Markov persuasion processes: Learning to persuade from scratch"

[Bernasconi et al. 2022] "Sequential information design: Learning to persuade in the dark"

[Cacciamani et al. 2023] "Online mechanism design for information acquisition"

[Azar et al. 2017] "Minimax regret bounds for reinforcement learning"

[Jin et al. 2020] "Provably efficient reinforcement learning with linear function approximation"

**Questions:**

See weaknesses above.

---

### Official Review · Reviewer_nJZN · 2024-10-27

**Soundness:** 3
**Presentation:** 2
**Contribution:** 2
**Rating:** 3
**Confidence:** 3

**Summary:**

The paper applies techniques from Bayesian persuasion frameworks to Markov games. It studies the problem in which an informed principal steers a set of agents to desired outcomes. First, it considers the case in which the feedback received from the principal is limited, i.e., the principal observes only the reward associated with the action recommended to the agents. Furthermore, the paper also studies a setting with additional feedback. The paper achieves tight regret guarantees in the case of bandit feedback and proves this tightness with a lower bound.

**Strengths:**

I think the main contribution of the paper is the setting itself. Indeed, information design, a.k.a. Bayesian persuasion, has received a lot of attention over the last few years. However, there are only a few works that consider how to apply this framework to Markovian settings.

**Weaknesses:**

Weaknesses:
- There are several points of confusion regarding the model considered in the paper. At the beginning, you state, "To bypass this challenge, we assume that the principal has the power to take actions on behalf of the agents, and she additionally suffers from the violation of the BCE constraint." However, then you say: "Based on the principal’s action and the recommended action, the agents take an action..." This poses the question of whether the principal is self-playing or if the agents play an active role in this scenario.

- I did not check the math in detail. However, my intuition is that the main algorithmic contribution lies in a sort of exploration (and planning) phase followed by a final commitment phase. Thus, while I acknowledge that it may not be straightforward to apply techniques from previous papers to this setting to perform these tasks, I also believe that the algorithmic approach is not particularly novel.

- Furthermore, it is well known that explore-and-commit approaches achieve $O(T^{2/3})$ regret and violation bounds. I believe that the main challenge is to study the tightness of such an algorithm for different values of $\alpha$, as done in [Bernasconi et al., 2022]. Therefore, at a given state, my evaluation is below the acceptance bar, given the high standards of ICLR.

Typos and minor weaknesses:
- It's up to the authors, but I think that the plot describing the interaction does not help in understanding the actual interaction. Indeed, why choose a 'circular' plot when the interaction is sequential?
- At line 193, a full stop is missing.
- Is in Eq. (1) the quantity $\nu_h - \nu_h$ correct?

**Questions:**

- Is it possible with your algorithm to achieve a tight upper bound on both constraint violations and regret with different values when $\alpha \ne 2/3$?
- Is the upper bound tight in the case of additional feedback? I am not sure you can use the same analysis in multi-armed bandits, as here you receive different feedback at each round.
- Are the agents myopic?

---

### Official Review · Reviewer_Ay4d · 2024-11-04

**Soundness:** 3
**Presentation:** 2
**Contribution:** 1
**Rating:** 3
**Confidence:** 3

**Summary:**

This paper studies information design in Markov games. The focus is on an online settings in which a principal aims at maximize his rewards. At the same time the principal must guarantee that the incentive compatibility constraints are only slightly violated. The paper provides a lower-bound on the regret/violations tradeoff that the learner can achieve. Then, the paper proposes  an algorithm that guarantees $T^{2/3}$ regret and violations. Finally, the authors shows that with additional feedback is it possible to achieve $\sqrt{T}$ regret and violations.

**Strengths:**

The paper studied a novel setting.

**Weaknesses:**

I find the technical novelty of the paper quite limited. Most results follows from an adaptation of previous works (e.g., Bernasconi et al. 2022) to MDPs.

**Questions:**

Which challenges arise in applying known techniques used in previous settings? The results look very similar.

---

### Official Review · Reviewer_EisR · 2024-11-13

**Soundness:** 2
**Presentation:** 1
**Contribution:** 3
**Rating:** 5
**Confidence:** 3

**Summary:**

This work studies an incentive and information design problem for a multi-agent Markov game. Specifically, this paper aims to find the optimal actions that the principal should take (incentive design) and the optimal actions/signals that the principal should recommend to agents (information design) such that the expected principal reward is maximized (regret) while agents have no incentive to deviate from the recommended actions (constraint violation). The authors propose an explore-then-commit type algorithm incorporating reward-free exploration. This paper provides theoretical analyses of the problem and the proposed algorithm.

**Strengths:**

- The dynamic incentive and information design (DIID) problem and framework studied seem interesting and can be used to model many applications.
- The proposed algorithm is straightforward and intuitive.

**Weaknesses:**

- Overall, this paper is not easy to follow.
- In the contribution paragraph and Theorem 4.1, how did it come about that $O(\cdot)$ is used for lower bound expression?
- There are so many notations in this framework. A notation table is needed to keep track.
- I would suggest moving section 5 to the appendix and adding a conclusion/discussion section to improve the clarity and readability of the paper.
- There is no empirical validation for the performance of the proposed algorithm

**Questions:**

- Could the authors discuss the limitations and future directions of this work?
- In Section 4.2, could the authors explain the reward-free exploration procedure they used in more detail? It seems adapting the standard reward-free exploration to this problem is one of the core algorithmic design contributions. However, it is not discussed sufficiently in the paper.

---

### Note · Authors · 2024-11-15

I have read and agree with the venue's withdrawal policy on behalf of myself and my co-authors.